PHILOSOPHY OF BIOLOGY

# The meanings of 'function' in biology and the problematic case of de novo gene emergence

**Abstract** The word function has many different meanings in molecular biology. Here we explore the use of this word (and derivatives like functional) in research papers about de novo gene birth. Based on an analysis of 20 abstracts we propose a simple lexicon that, we believe, will help scientists and philosophers discuss the meaning of function more clearly.
DOI: https://doi.org/10.7554/eLife.47014.001

DIANE MARIE KEELING, PATRICIA GARZA, CHARISSE MICHELLE NARTEY* AND ANNE-RUXANDRA CARVUNIS*

*For correspondence: charisse. crenshaw@gmail.com (CMN); anc201@pitt.edu (A-RC)

**Competing interests:** The authors declare that no competing interests exist.

## Introduction

The scientific enterprise is inherently rhetorical (*Condit, 1999*; *Fahnestock, 2002*; *Ceccarelli, 2001*). When we describe observations, articulate hypotheses or develop theories, we consciously or unconsciously select words to convey the meaning of our work. These words, rather than our own understanding, are what is published, read, interpreted and possibly built upon for the years to follow. Language influences how we communicate, how we think, and how we practice science. A specific and contextual use of language is therefore paramount to a productive global scientific endeavor. In practice, however, confusion and even conflict can arise when the same word is understood differently by authors and readers.

A reader's interpretation of a word in a text depends on who the reader is. What is their field of training? In what country did they grow up? In what decade were they born? What are their metaphysical presuppositions? All these and countless other factors act as a filter between the words themselves and the meanings that readers assign to them. These factors are out of the author's control, and their effect can be amplified when the word is part of everyday language.

A striking example of different interpretations of words leading to controversy was the debate, prompted by the first papers from the ENCODE Project, about what fraction of the human genome is 'functional'. Is it approximately 80%, as suggested by biochemical evidence, or is it closer to 10%, as it appears from evolutionary evidence (*The ENCODE Project Consortium, 2012*; *Doolittle, 2013*; *Graur et al., 2013*; *Kellis et al., 2014*)?

Of course, the answer to this question entirely depends on what exactly is meant by the word 'function'. For most evolutionary biologists, function relates to selection (that is, the effect for which the gene was selected in the past at the organismal level). For most molecular geneticists and biochemists, it is generally associated with a molecule's activity (such as catalysis or binding), independent of the historical factors that led to its existence. These meanings are epistemological constructions influenced by scientific training. In everyday language, the word 'function' is associated with many additional concepts ranging from someone's professional

occupation to the purpose for which a machine has been engineered. Yet some argue that the evolutionary meaning of function is the only one that is relevant for DNA sequences (*Graur et al., 2013*) and that when referring to other meanings scientists should use other words such as 'effect' or 'activity' (*Doolittle et al., 2014*).

This position, and the arguments against it, echo an ongoing conversation in the philosophy of science (*Cummins, 1975*; *Millikan, 1989*; *Neander, 1991*; *Amundson and Lauder, 1994*; *Garson, 2011*; *Griffiths, 2009*). In brief, the conversation divides those who argue function should mean *why an entity does what it does* (the selected effect definition of function) and those who argue it can also mean *what an entity does* (the causal role definition of function) (*Laubichler et al., 2015*). Function strictly defined as selected effect is the historical explanation for the existence of an entity (*Millikan, 1989*). Function as causal role is ahistorical and describes the contribution of an entity to a complex system (*Cummins, 1975*). Other theories of biological function have been proposed (*Wouters, 2003*; *Mossio et al., 2009*; *Roux, 2014*), but the selected effect/causal role perspectives have dominated the conversation in the context of genomics.

Far from a fruitless dispute over semantics, the rhetorico-scientific debate has sparked a number of thoughtful studies that have advanced thinking at the interface of evolutionary biology and genomics. For instance, one study interrogates the relationship between organismal complexity and the number of functional elements in a genome (*Doolittle, 2013*), another highlights the discrepancies between different lines of evidence used to infer functionality of DNA loci (*Kellis et al., 2014*), and another refines the evolutionary classifications of genomic functions (*Graur et al., 2015*). However, only a small fraction of biologists – mostly evolutionary biologists – are aware of the rhetorical context. Most scientific publications do not explicitly include a definition of 'function' to clarify the meaning of the reported findings.

It can be argued that this imprecision has tangible implications for the scientific practice. For better or worse, in biological research there is a conflation between what is 'functional' and what 'matters'. Only those genomic sequences deemed 'functional' are worthy of being curated by reference databases, of being named, cloned

or incorporated into grant proposals for mechanistic studies. Today, scientists cannot agree on the number of functional genes in the human genome (*Pertea et al., 2018*; *Jungreis et al., 2018*), or on what evidence should be required to elevate noncoding RNAs from mere transcriptional noise to functional regulatory elements (*Doolittle, 2018*). The general confusion about what exactly is meant by 'function' across the literature is such that some have pleaded for the community to deal with the 'F-word' urgently (*Doolittle et al., 2014*; *Doolittle, 2018*).

As an interdisciplinary group of scholars, we sought to understand to what extent the existence of differing meanings of 'function' actually impairs the scientific enterprise. We focused our attention on a relatively recent subfield of evolutionary genomics studying the specific case of de novo gene birth. This field attempts to understand how new 'functional' genes can emerge without having derived from another gene as their ancestor. The process of de novo gene birth was long thought to be implausible on the basis that, as Francois Jacob wrote: "the probability that a functional protein would appear de novo by random association of amino acids is practically zero" (*Jacob, 1977*). However, the explosion of new genome sequences has revealed that many genes have species- or lineage-specific sequences (*Khalturin et al., 2009*), suggesting that they lack gene ancestors. Studies of a growing number of individual gene candidates have confirmed their de novo emergence, fueling many genomic and evolutionary studies to evaluate the scale and mechanisms of the de novo gene birth phenomenon (*Tautz and Domazet-Lošo, 2011*; *McLysaght and Hurst, 2016*; *Van Oss and Carvunis, 2019*).

The meanings of function are at the heart of what constitutes a de novo gene birth event. For a genomic sequence to be labelled as a gene, it must by definition have a function; it must express a product that participates in cellular processes and affects phenotypes in a way that is being maintained by selection. If such a gene has evolved de novo, the locus it came from by definition was not a gene, thus did not have a function, or at least not a function of the same nature as the one the new gene has. The molecular objects of study are thus transitioning between a state without a function and a state with a function. They cannot have upon birth a

function by the strict selected effect definition (*Millikan, 1989*), since their existence cannot have been caused historically by a past selection. The transformative nature of the de novo emergence process thus renders the debates about when and how a locus actually becomes functional highly contentious.

Let us consider a hypothetical example to illustrate the difficulty of thinking about the function of recently emerged coding elements. Imagine a locus that has become transcribed and translated for the first time in somebody's gamete, generating a novel protein whose expression propagates to future generations. The corresponding protein has no activity whatsoever. Does this locus correspond to a de novo functional gene? Not according to the selected effect definition, since there is no past selection to explain why the locus is here. Not according to the causal role definition either, since it does not do anything. What if the protein happens to confer a fitness benefit to the organism? Still, the locus would not have a function according to the strict selected effect definition (*Millikan, 1989*), although it might be in the process of acquiring one. What if the protein causes a deadly disease? In that case many may find it acceptable to use derivatives of function and write 'the locus functions in the development of disease', or 'the locus is functional', but not 'the function of the locus is to cause a deadly disease' (*Doolittle et al., 2014*). One might cautiously call this locus a proto-gene (*Carvunis et al., 2012*), Zombie or Lazarus DNA (*Graur et al., 2015*), rather than a functional gene. But, at any rate, this locus 'matters'

(*Ardern, 2018*) and there is a pressing need to find the right words to express what it does, why and how.

In addition to these fundamental considerations, the de novo field is interdisciplinary and relatively young. It is therefore rich in diverse perspectives and trainings, and de novo researchers lack a commonly accepted jargon. All these challenges make de novo gene evolution a well-suited test bed to evaluate what meanings of function are circulating in this field and whether, and to what extent, the meanings of function hinder scientific communication.

## Constructing a model of function for de novo gene birth research

We sought to construct an understanding of function specifically tailored to de novo gene birth. We reasoned that this aim would be best achieved by studying how the term is used in the scientific practice of this particular field of research. Indeed, the objects of study and the technical methodologies in this field may lend themselves to different interpretations of function than in other fields such as regulatory genomics, physiology or ecology. In order to derive an initial model of function adapted to de novo gene birth research, we first rhetorically analyzed the scientific literature in the field together with philosophical publications about genomic function. We then applied the constant comparative method of the grounded theory of social sciences (*Glaser and Strauss, 1967*) to samples of 20 published abstracts in the field (see Methods). Through an organized, iterative process of defining and discussing usages of the term, we

**Table 1.** The Pittsburgh model of function.
The hierarchical order of the meanings did not directly derive from our textual analysis, but was inspired from a reductionist interpretation of the flow of genetic information over time and space. It also reflects a possible ordering of the series of properties that must be acquired by a locus to undergo de novo gene birth.

| Meanings | Definitions |
|---|---|
| Evolutionary Implications | The object's influence on population dynamics over successive generations, as enabled by its physiological implications and their interplay with environmental pressures |
| Physiological Implications | The object's involvement in biological processes as enabled by a set of its capacities, interactions and expression patterns, independent of cross-generational considerations |
| Interactions | Physical contacts, direct or indirect, between the object under investigation and the other components of a system, including contacts that mediate chemical transformations |
| Capacities | Intrinsic physical properties of the object under investigation; the necessity of the object's behavior given an environment (eg., structural constraints) |
| Expression | The presence or amount of the object under investigation (RNA or protein object), or the presence or amount of its transcription or translation products (DNA object) |
| Vague | Sufficient evidence was not found to infer one or more meanings of function within this model, nor to derive a new meaning |

DOI: https://doi.org/10.7554/eLife.47014.002

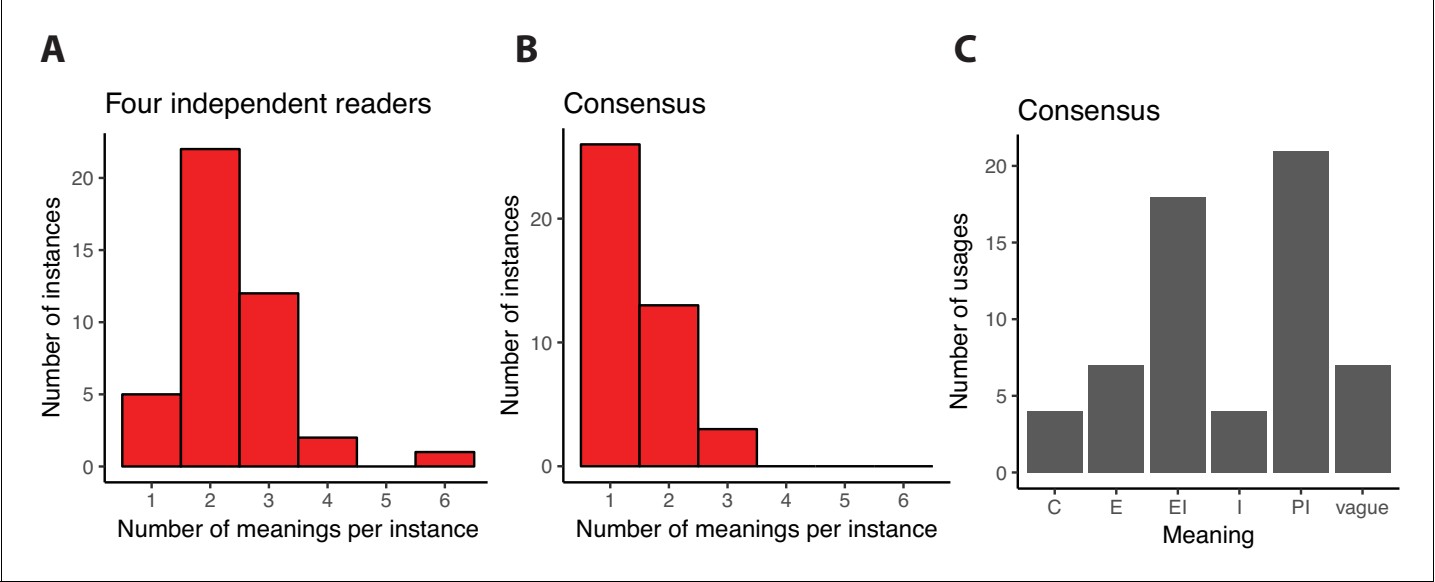

**Figure 1.** Interpreting the word function in scientific abstracts related to de novo gene birth. We analyzed a sample of 20 abstracts containing 42 instances where the word function or one of its derivatives was used to describe DNA, RNA or protein objects. First, each of us read the abstracts independently and assigned one or several of the meanings of function as defined in the Pittsburgh model to each of these instances. The distribution of the number of distinct meanings that we assigned to the 42 instances is shown in panel (A). For only 5 instances did all of us independently assign the same unique meaning, suggesting that function is most often interpreted in multiple ways by independent readers. Next, we discussed each instance to see if we could reach consensus assignments based on the textual evidence. Consensus was built through conversations and agreement between the readers, rather than majority opinion. The distribution of the number of unique meanings assigned after consensus agreement to each of the 42 instances is shown in panel (B). Most (26/42) instances are now assigned to a single meaning. When more than one meaning remains, the readers agreed that the textual evidence supported multiple meanings except for one instance where consensus could not be reached and three meanings were assigned to reflect all the differing interpretations of our team members. In panel C, we show the number of times each of the five meanings of function defined in the Pittsburgh model is assigned to an instance of function.

DOI: https://doi.org/10.7554/eLife.47014.003

The following source data is available for figure 1:

**Source data 1.** Independent and consensus assignments.
DOI: https://doi.org/10.7554/eLife.47014.004

inductively converged on the interpretation that, in this set of abstracts, authors writing about the function of a molecular object were almost always describing one or more of the following properties of the object: Expression, Capacities, Interactions, Physiological Implications and Evolutionary Implications. These properties represent five meanings of function that are defined in *Table 1*.

Conveniently, these five meanings map to an interpretation of the epistemological flow of genetic information over time and space. Starting from an object's presence (Expression), we consider its physical properties (Capacities), binding partners within a system (Interactions), phenotypic impact (Physiological Implications) and influence on population dynamics (Evolutionary Implications). Accordingly, we propose to relate these five meanings of function in a hierarchy inspired from molecular, evolutionary

and systems biology (*Noble, 2006*; *Medina, 2005*; *Ernst and Carvunis, 2018*). This hierarchy reflects a possible ordering of the series of properties that must be acquired by a locus to undergo de novo gene birth. Altogether, the definitions and hierarchical organization are hereafter referred to as the Pittsburgh model of function (*Table 1*). This model summarizes our analyses of how the term function is used in the field. The model also includes a sixth category labelled 'vague', for the few instances where we could neither assign any of the five meanings, nor infer a sixth meaning from the context.

Like the molecules they describe, the five meanings of function are interrelated in complex bottom-up and top-down ways that complicate causal inferences (*Noble, 2006*). For instance, as has been discussed in the context of the ENCODE debate, Expression is not sufficient to

cause Evolutionary Implications (*Doolittle, 2018*). Inversely, Evolutionary Implications do not necessarily imply Expression since a locus can influence population dynamics through a DNA regulatory activity. The methodological details of the study determine whether the burden of proof has been met to assign one or several of the proposed five meanings of function to a molecular object. Such rigor in functional inference is especially critical for the field of de novo gene birth, where the objects of interest often display some but not all of the properties of established genes (*McLysaght and Hurst, 2016*; *Carvunis et al., 2012*; *Ruiz-Orera et al., 2018*). Our model acknowledges epistemological relationships between different meanings of function while enabling researchers to describe them independently of each other.

This model presents practical advantages relative to pre-existing ones because it is tailored to one field of research. In particular, it differentiates between different types of biochemical activities for de novo emerging sequences; this enables scientists to articulate more specific functional inferences than with broad terms that generalize across fields such as 'causal role' or 'mere activities' (*Cummins, 1975*; *Wouters, 2003*; *Doolittle, 2013*). Rather than theorizing upon the legitimacy of what function should mean, our model decomposes this complex concept into a hierarchical organization of measurable properties of the object in time and space according to meanings that are circulating in the field currently.

The Pittsburgh model of function can be seen as a conceptual tool well adapted to describe molecular objects (DNA loci, RNA transcripts and proteins) in a manner that roughly maps to subfields of training and associated measurement techniques currently used in de novo gene birth research. For instance, while RNA-sequencing readily informs upon Expression and yeast two-hybrid upon Interactions, discovering interactions by yeast two-hybrid does not allow the researcher to conclude that a locus is natively expressed; so too, establishing that a locus is expressed does not imply that the corresponding product necessarily interacts with other elements in the cell. Evidently, neither technique gives direct insights into Evolutionary Implications since Expression and Interactions are necessary but not sufficient to cause Physiological Implications, let alone Evolutionary Implications. Separating these meanings from one another enables communicating with increased precision about what the findings are, thereby helping to

fallacious logical shortcuts such as 'this protein is expressed therefore it is functional therefore it is under selection'.

The meanings of Physiological Implications and Evolutionary Implications are deliberately broad to allow for detailed descriptions beyond the somewhat restrictive notion of selected effect. For instance, Evolutionary Implications can refer to selection upon a trait driven by the object in the past, as in the strict definition of selected effect (*Millikan, 1989*), but it can equally describe other ways the object may influence population dynamics such as runaway selection or novel fitness enhancing effects.

The Pittsburgh model has important limitations. Tested on only 20 abstracts related to the single field of de novo gene birth, by no means does it represent a complete ontology of function. Constructed in a field-specific manner, our model may or may not extend to genomic elements outside of the gene birth framework, such as selfish elements or promoters. It cannot be readily applied to biological objects other than DNA, RNA and protein objects.

There may be additional meanings available beyond the five we included, and different ways to characterize the relationships between these meanings. Function can for instance refer to an object's influence on ecological behavior, but this was not the case in the small sample of abstracts we analyzed. Interactions may be better placed below Expression in a hierarchical model that would be tailored to regulatory genomics, where the focus is on how physical interaction of DNA elements with diverse proteins determine regulatory outputs, rather than for de novo gene birth, where the focus is on the loci being expressed themselves. As well, Physiological Implications and Evolutionary Implications may be further subdivided to account for the diverse relationships evident within these concepts and the methodological procedures that inform them.

Lastly, our model does not attempt to resolve the teleological dimension of function, especially as it relates to physiological processes and an object's involvement therein (*Roux, 2014*). Despite these limitations, our proposed model provides a practical lexicon that maps to prevalent data generation and interpretation practices (*Laubichler et al., 2015*). Therefore, we hope that our work will help researchers from different disciplines with differing backgrounds communicate more effectively about de novo gene birth.

## How multiple meanings of function are used in the field of de novo gene birth

With this model in hand, we analyzed whether the multiple meanings of function impact understanding of the literature in the field of de novo gene birth. Each member of our team independently assigned one or more meanings to each instance of the word function found in our database of abstracts, gathering contextual evidence primarily from the sentence in which the term was used (4 members; 20 abstracts; 42 instances including 25 nouns, 12 adjectives, 3 verbs and 2 adverbs; see Methods). The abstracts generally provided enough context for each independent reader to confidently assign meanings to most instances, with only rare assignments of the label 'vague' (9 instances with one or more vague assignment, none unanimous). Hence, the Pittsburgh model gave readers a key to decipher more specifically what authors meant by the word function. However, and importantly, a quantitative analysis revealed only that 12% (5/42) of assignments were unanimous, where the same single meaning was chosen by all readers (*Figure 1A*). In other words, the same instance of the word in the same sentence was most of the times (37/42 = 88%) interpreted differently by our team members. The most confusing instances, with 4 or 6 distinct assignments, were all assigned 'vague' by at least one reader. This analysis indicates that when the meaning of function is unspecified, the literature in this field of research can become confusing.

Two mechanisms could in theory explain why our independent textual analysis led to multiple meanings being assigned to the same instance of function in 88% of cases. On the one hand, it could be that the different readers often interpreted the same text differently due to their different backgrounds. On the other hand, the word function may often be used by authors to reflect several of the five meanings in our model simultaneously.

To determine to what extent each of these mechanisms was responsible for our observations (*Figure 1A*), we collectively reviewed and discussed all independent assignments for each instance of the word and built a consensus set of assignments. When we agreed that the textual evidence supported more than one meaning for a single instance, we assigned multiple meanings to this instance as a consensus. This consensus is thought to best reflect what was intended by the authors of the abstracts, although we cannot know for sure without interviewing the authors.

Consensus was successfully reached for all but one instance, suggesting that the Pittsburgh model enabled adequate description of most of the instances of function in our abstract database. We then quantified again the fraction of instances of function that were assigned a unique meaning. This number grew from 5/42 (12%) in the original independent assignments (*Figure 1A*) to 26/42 (62%) in the consensus assignments (*Figure 1B*). This shows that, in our analyses, 21/42 (50%) instances were interpreted differently by at least one of four readers upon independent reading of the texts.

It is notable that, after establishing consensus assignments, still 16/42 (38%) instances were assigned two or three meanings (*Figure 1B*). This suggests that this one word is often used to mean different things, and that the plurality of function cannot be easily disentangled.

The meanings most frequently found in our consensus assignments were Physiological Implications (21 instances) and Evolutionary Implications (18 instances; *Figure 1C*). These two were also most often found assigned together in cases where a single instance was assigned two or three meanings (6 instances). We did not observe strong differences in how function was interpreted when it was used as a noun or an adjective, but such differences may become detectable in a larger text sample. Interestingly though, all but one instances with a vague consensus assignment were nouns. The extent of plurality (*Figure 1A and B*), as well as the distribution of meanings (*Figure 1C*), are likely to be field-specific. Yet our rhetorical approach leads us to conclude that the use of function in this field is hard to interpret, and that further nuance in writing would assist the reader in understanding how the authors intend their results to be interpreted.

## Interpretation and recommendations

In summary, our results provide quantitative support to previous assertions (*Doolittle, 2018*; *Laubichler et al., 2015*) that the word function carries convoluted meanings that complicate the scientific conversation. Our rhetorical analysis shows that function becomes an ambiguous concept when applied to the edge case of de novo gene birth, which can lead to confusion in interpretation of the literature. Whether multiple meanings are intended by the authors, or

meanings unintended by the authors are interpreted by the readers, or a combination of both, the fact is that scientific communication is hindered by the use of this word within the de novo gene birth field. There may be excellent theoretical arguments to be made about why function *should* mean one thing or another, but the cultural diversity of readers in this emerging field effectively prevents a unique definition to be imposed in practice. We believe that it will be productive for scientists to acknowledge the diversity of onto-epistemological perspectives in this field and adapt their writing style accordingly. Rather than privileging one meaning of function over another, we endorse qualifying the use of function or avoiding the word altogether (*Doolittle, 2018*). We hope our model will provide a useful tool for scientists to contextualize their writing so the relationship between the observations reported and the functional inferences made can be clarified and the risk of misunderstanding can be reduced.

Function is a concept that depends on the methodological practices, measurement procedures and habits of scientists. As these change, so too will the concept of function change and adapt to specific subfields of research, because it is contingent and always in a state of flux. Function requires ongoing attention and theorizing by the scientific community. Therefore, we expect our model to develop and adapt with scientific practice. We encourage researchers to refine the model, adapt it to other subfields of genomics and other types of biological objects (metabolites, cell types, organs), and propose alternatives. One can envision, for instance, exploiting the power of computational natural language processing to generate field-specific models by automatic analysis of large bodies of literature (*Friedman et al., 2001*; *Groth et al., 2016*), perhaps using our modest manual analysis as a training set. More generally, we hope that interdisciplinary conversations about philosophy (*Laplane et al., 2019*), rhetoric and scientific concepts will accompany the emergence of new scientific fields in the future.

## Methods

### An interdisciplinary mixed methods approach

We approached the question of the meanings of function in the de novo gene birth literature using a mixed methods study design adapted from rhetorical studies and applied communication (*Creswell, 2014*; *McGreavy et al., 2015*; *Dewulf et al., 2007*; *Thompson, 2009*). First and throughout, we performed a rhetorical analysis by interrogating the assumptions made by scientists within this field and how they inform language in the published literature (*Gross, 1990*). In addition, we performed a qualitative analysis to iteratively build a model of function as it applies to the field (*Strauss and Corbin, 1990*). Finally, we performed a quantitative content analysis to analyze how the multiple meanings of function affect understanding of the literature in the field (*Neuendorf, 2016*).

### Paper selection

A library of 20 published papers that included the term function or its derivatives (functional, functioning) in the abstract was assembled by a team member who is also a published expert in the field of de novo gene birth (*Table 2*). Publication dates span from 1992 to 2017, with most dated after 2012 because this is a recently expanding field. The papers were chosen to span a variety of journals, countries, citation counts, model organisms, methodologies and scope, in order to derive a context specific rhetorical argument (*McGee, 1990*). This library is estimated to represent ~2% of the literature published on the topic of de novo gene emergence, as a Google Scholar search returns 972 results in December 2018 (' ''de novo gene birth'' OR ''de novo gene evolution'' OR ''de novo gene emergence'' OR ''de novo genes'' ').

### Instance selection

Instances of the use of the word function, or its derivatives, were selected for analysis because they explicitly related to a DNA, RNA or protein object within a sentence of the abstracts. We focused on abstracts, as they present a self-contained statement of the motivations, results and conclusions of the studies and they are the text seen by most readers. Instances within article titles were not considered, neither were those where function was used as a subject, referring to bioprocesses such as: *We then introduce recent findings that have opened a path to the study of the evolution of novel functions and pathways via novel genes* (*Ding et al., 2012*). Forty-two usages (25 nouns, 12 adjectives, 3 verbs and 2 adverbs) were analyzed across 20 abstracts.

**Table 2.** References for 20 abstracts analyzed in our study.
Countries (based on affiliations of all authors) and model organisms are included to display the diversity of the abstracts.

| Papers | Countries | Model Organisms |
|---|---|---|
| Keese, P. K., and Gibbs, A. (1992). Origins of genes: 'big bang' or continuous creation? *PNAS* **89**:9489–9493. | Australia | Cellular life, Viruses |
| Kastenmayer, J. P., Ni, L., Chu, A., Kitchen, L. E., Au, W. C., Yang, H.,. .. and Basrai, M. A. (2006). Functional genomics of genes with small open reading frames (sORFs) in *S. cerevisiae*. *Genome Research* **16**:365–373. | USA | *S. cerevisiae* |
| Levine, M. T., Jones, C. D., Kern, A. D., Lindfors, H. A., and Begun, D. J. (2006). Novel genes derived from noncoding DNA in *Drosophila melanogaster* are frequently X-linked and exhibit testis-biased expression. *PNAS* **103**:9935–9939. | USA | *D. melanogaster* |
| Stepanov, V. G., and Fox, G. E. (2007). Stress-driven in vivo selection of a functional mini-gene from a randomized DNA library expressing combinatorial peptides in *Escherichia coli*. *Molecular Biology and Evolution* **24**:1480–1491. | USA | *E. coli* |
| Cai, J., Zhao, R., Jiang, H., and Wang, W. (2008). De novo origination of a new protein-coding gene in *Saccharomyces cerevisiae*. *Genetics* **179**:487–496. | China | *S. cerevisiae* |
| Zhou, Q., Zhang, G., Zhang, Y., Xu, S., Zhao, R., Zhan, Z.,. .. and Wang, W. (2008). On the origin of new genes in *Drosophila*. *Genome Research* **18**:1446–1455. | China | *Drosophila* |
| Xiao, W., Liu, H., Li, Y., Li, X., Xu, C., Long, M., and Wang, S. (2009). A rice gene of de novo origin negatively regulates pathogen-induced defense response. *PLoS One* **4**:e4603. | China, USA | rice |
| Carvunis, A. R., Rolland, T., Wapinski, I., Calderwood, M. A., Yildirim, M. A., Simonis, N.,. ..and Vidal M. (2012). Proto-genes and de novo gene birth. *Nature* **487**:370–374. | Belgium, France, USA | *S. cerevisiae* |
| Ding, Y., Zhou, Q., and Wang, W. (2012). Origins of new genes and evolution of their novel functions. *Annual Review of Ecology, Evolution, and Systematics* **43**:345–363. | China, USA | |
| Tautz, D., Neme, R., and Domazet-Lošo, T. (2013). Evolutionary Origin of Orphan Genes. In: *Encyclopedia of Life Sciences*. John Wiley & Sons. DOI: https://doi.org/10.1002/9780470015902.a0024601 | Croatia, Germany | |
| Reinhardt, J. A., Wanjiru, B. M., Brant, A. T., Saelao, P., Begun, D. J., and Jones, C. D. (2013). De novo ORFs in *Drosophila* are important to organismal fitness and evolved rapidly from previously non-coding sequences. *PLoS Genetics* **9**:e1003860. | USA | *D. melanogaster* |
| Wissler, L., Gadau, J., Simola, D. F., Helmkampf, M., and Bornberg-Bauer, E. (2013). Mechanisms and dynamics of orphan gene emergence in insect genomes. *Genome Biology and Evolution* **5**:439–455. | Germany, USA | Insects |
| Brylinski, M. (2013). Exploring the 'dark matter' of a mammalian proteome by protein structure and function modeling. *Proteome Science* **11**:47. | USA | *M. musculus* |
| Li, D., Yan, Z., Lu, L., Jiang, H., and Wang, W. (2014). Pleiotropy of the de novo-originated gene MDF1. *Scientific Reports* **4**:7280. | China | *S. cerevisiae* |
| Wirthlin, M., Lovell, P. V., Jarvis, E. D., and Mello, C. V. (2014). Comparative genomics reveals molecular features unique to the songbird lineage. *BMC Genomics* **15**:1082. | USA | Songbirds |
| Suenaga, Y., Islam, S. R., Alagu, J., Kaneko, Y., Kato, M., Tanaka, Y.,. .. and Nakagawara, A.(2014). NCYM, a Cis-antisense gene of MYCN, encodes a de novo evolved protein that inhibits GSK3β resulting in the stabilization of MYCN in human neuroblastomas. *PLoS Genetics* **10**:e1003996. | Japan | Human |
| Arendsee, Z. W., Li, L., and Wurtele, E. S. (2014). Coming of age: Orphan genes in plants. *Trends in Plant Science* **19**:698–708. | USA | *A. thaliana* |
| Ruiz-Orera, J., Hernandez-Rodriguez, J., Chiva, C., Sabidó, E., Kondova, I., Bontrop, R.,. .. and Albà, M. M. (2015). Origins of de novo genes in human and chimpanzee. *PLoS Genetics* **11**:e1005721. | Spain, The Netherlands | Human, Chimpanzee |
| Couso, J. P., and Patraquim, P. (2017). Classification and function of small open reading frames. *Nature Reviews Molecular Cell Biology* **18**:575–589. | Spain, UK | *D. melanogaster* |
| Luis Villanueva-Cañas, J., Ruiz-Orera, J., Agea, M. I., Gallo, M., Andreu, D., and Albà, M. M. (2017). New genes and functional innovation in mammals. *Genome Biology and Evolution* **9**:1886–1900. | Spain | Mammals |

DOI: https://doi.org/10.7554/eLife.47014.005

### Qualitative analysis and iterative model construction

#### The need for an improved model of function

We began the qualitative analysis by establishing the need for refining the selected effect/causal role binary model discussed in the philosophical literature of genomic function. First, the model has contentious philosophical implications, in particular as they relate to teleology, that have been extensively discussed (*Allen and Bekoff, 1995*; *Manning, 1997*; *Buller, 2001*; *Roux, 2014*). Second, the epistemological reduction of function into a dichotomy

**Table 3.** Examples of each meaning of function as assigned to instances of usage.
Underlined portions of sentences serve as the contextual evidence used to assign the 'code', or meaning, to the bolded instances analyzed.

| Reference | Instance of function usage | Consensus meanings |
|---|---|---|
| Wirthin et al., 2014 | 'Here we performed a comparative analysis of 48 avian genomes to identify genomic features that are unique to songbirds, as well as an initial assessment of **function** by investigating their <u>tissue distribution</u> and predicted <u>protein domain structure</u>.' | Expression, Capacities |
| Brylinski, 2013 | A subsequent structure-based **function** annotation of small protein models exposes 178,745 putative <u>protein-protein interactions</u> with the remaining gene products in the mouse proteome, 1,100 potential <u>binding sites for small organic molecules and 987 metal-binding signatures</u>. | Interaction |
| Li et al., 2014 | 'Therefore, MDF1 **functions** in two important molecular pathways, <u>mating and fermentation</u>, and mediates the crosstalk between <u>reproduction</u> and <u>vegetative growth</u>.' | Physiological Implications |
| Ruiz-Orera et al., 2015 | 'In general, these transcripts show little evidence of <u>purifying selection</u>, suggesting that many of them are not **functional**' | Evolutionary Implications |

DOI: https://doi.org/10.7554/eLife.47014.006

mutes the complex ontological relationships between the multiple ways the term can be used and, problematically, leaves theoretical assumptions and measurement constraints implicit and unspecified (*Laubichler et al., 2015*). Third, evidence suggests that the model has not been widely adopted by the scientists publishing in the field of de novo gene birth. For example, a Google Scholar search for ' [''de novo gene birth'' OR ''de novo gene evolution'' OR ''de novo gene emergence'' OR ''de novo genes''] AND ''causal role'' ' yields only 10 results in March 2019, whereas 1050 results are found when the AND clause is lacking.

## Preliminary model construction

We reasoned that it might be possible to construct a novel model of function by studying the specific uses of the word in the context of the scientific discourse about de novo evolving molecules. We thus began a series of philosophical conversations moderated by a member of our team who is a published expert in interdisciplinary rhetoric and teaches collaborative problem solving. Our objectives were three-fold: i) to reduce teleological overtones; ii) to increase the focus on how ontological relationships map to ongoing practices in biological research; and iii) to propose alternate terms that could conveniently be adopted by scientists. These conversations resulted in the construction of a preliminary theoretical model.

## Model refinement

Next, we evaluated the accuracy of our preliminary model by conducting a content analysis of the use of function in a sample of the 20 abstracts in our library (*Neuendorf, 2016*). Individually, we interpreted the meaning of function using the context of the sentence containing the word first, and the general context of the abstract second, to attempt to assign one of the definitions from our preliminary model to each instance. Throughout this process, we identified inconsistencies between our preliminary model and the actual usages of function in the texts, leading to further refinements of the model. This process was repeated iteratively on samples consisting of up to 17 of the 20 papers in our library, until a reasonable agreement between theory and texts was reached and agreed upon by each member of our team (*Neuendorf, 2016*). The model that emerged from this iterative work was validated using the remaining three texts in the library. This methodology of iterative model construction is known in the social sciences as the constant comparative method of the grounded theory (*Glaser and Strauss, 1967*). This work resulted in a structured classification of the meanings of function specifically adapted to the de novo gene birth literature, which we named the Pittsburgh model of function after the geographical location where the model crystallized at the occasion of a collaborative retreat between our team members.

## Quantitative analysis

We used the Pittsburgh model of function to analyze whether the unspecified multiple meanings of function hinder understanding of the literature in the field of de novo gene birth. If we observed that independent readers tend to agree on which meaning was meant by the authors most of the time the term function is

used, we would conclude that the multiple meanings of function are *not* hindering communication in the field. If, in contrast, independent readers were found to frequently interpret the same instance of function in the same sentence differently, we would conclude that the unspecified use of function leads readers to misunderstand the literature.

We proceeded to perform a quantitative content analysis of the 42 usages of the word function found in our 20 abstracts by following commonly used social scientific guidelines (*Neuendorf, 2016*). First, each member of the team ('coders') read each of the 20 abstracts independently and assigned ('coded') the relationship they interpreted between the instance and at minimum one of the meanings of function from our model. Second, discordant assignments were discussed as a group and a consensus assignment was made that took into account all perspectives.

The coding rules were defined as follows:

- coding must only occur after the coder has read the entire title and abstract
- assignments must reflect what the coder thinks that the author meant given the context of the sentence and abstract, rather than what the coder thinks the scientific data presented actually demonstrates
- assignments must preferentially derive from the sentence in which the term is used alone
- if a meaning of function is described in the sentence, for example through an adjective or an adverb, this is the meaning that must be assigned
- assignments should take context into account when a meaning cannot be deduced by the sentence in which the term is used alone; in these cases, the relevant contextual evidence must be clearly highlighted
- when a single instance of the term is interpreted as multiple meanings, and context does not help distinguish them, then the meaning that is furthest in the progression from Expression to Evolutionary Implications must be coded (*Table 1*)

Examples of function usage and consensus meanings assigned are shown in *Table 3*. The entire data set of independent and consensus assignments is available in *Figure 1—source data 1*.

## Acknowledgements

The authors acknowledge the Association for the Rhetoric of Science, Technology, and Medicine (ARSTM) Pre-conference on Rhetorics of Resilience, hosted at the 2018 Biennial Conference of the Rhetoric Society of America. The authors also thank Dr William Bechtel and Dr Paul Nelson for useful conversations.

**Diane Marie Keeling** is in the Department of Communication Studies, College of Arts & Sciences, University of San Diego, San Diego, United States
 https://orcid.org/0000-0003-3613-9162

**Patricia Garza** is in the Colegio de Saberes, Mexico City, Mexico

**Charisse Michelle Nartey** is in the Department of Biological Sciences, University of Texas at Dallas, Richardson, United States
charisse.crenshaw@gmail.com
 https://orcid.org/0000-0001-5275-366X

**Anne-Ruxandra Carvunis** in the Department of Computational and Systems Biology and the Pittsburgh Center for Evolutionary Biology and Medicine, University of Pittsburgh School of Medicine, Pittsburgh, United States
anc201@pitt.edu
 https://orcid.org/0000-0002-6474-6413

*Author contributions:* Diane Marie Keeling, Conceptualization, Data curation, Formal analysis, Investigation, Methodology, Writing—original draft, Writing—review and editing; Patricia Garza, Conceptualization, Data curation, Formal analysis, Investigation, Methodology, Writing—review and editing; Charisse Michelle Nartey, Conceptualization, Formal analysis, Data curation, Investigation, Methodology, Writing—review and editing; Anne-Ruxandra Carvunis, Conceptualization, Data curation, Formal analysis, Funding acquisition, Investigation, Methodology, Writing—original draft, Writing—review and editing

*Competing interests:* The authors declare that no competing interests exist.

## Funding

| Funder | Grant reference number | Author |
|---|---|---|
| National Institutes of Health | R00GM108865 | Anne-Ruxandra Carvunis |
| Kinship Foundation | Searle Scholars Program | Anne-Ruxandra Carvunis |

The funders had no role in study design, data collection and interpretation, or the decision to submit the work for publication.

**Decision letter and Author response**
Decision letter https://doi.org/10.7554/eLife.47014.010
Author response https://doi.org/10.7554/eLife.47014.011

# Additional files

## Supplementary files
• Transparent reporting form DOI: https://doi.org/10.7554/eLife.47014.007

## Data availability
All data analyzed in this manuscript consists of published research abstracts that are freely available online. Source data file has been provided for Figure 1.

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
