## [Decision Letter]

Thank you for submitting your article "The meanings of "function" in biology and the problematic case of de novo evolution" for consideration by *eLife*. Your article has been reviewed by four peer reviewers, and the evaluation has been overseen by the *eLife* Features Editor. The following individuals involved in review of your submission have agreed to reveal their identity: Ford Doolittle (Reviewer #1); Etienne Roux (Reviewer #3); Lauren Cagle (Reviewer #4).

We would like to invite you to submit a revised manuscript that addresses the points raised by the referees (please see below).

Regarding point 11 from reviewer #3: please address this point by revising your manuscript so as not to present the second half of your work as a test of a hypothesis or model (rather than, for example, performing further work on other abstracts in order to attempt test your hypothesis).

Reviewer #1:

Introduction. The first sentence of this manuscript makes a very strong claim – not just that scientists inevitably express results in words, but that they do so "in terms intended to persuade or impress". At least that's how the OED in my Mac defines "rhetorical" and it's what scientists try NOT to do, when describing observations at least. Perhaps we never succeed in being "objective", and of course we do want to present theories in the best light, but the goal of science is NOT to "intend to persuade" in presenting data. Do authors mean to claim we never succeed in getting rid of such an intent? The second paragraph reminds us, usefully, that words mean different things to different people, which I grant – just not that any use of words is necessarily "rhetorical".

Introduction. I'm a little unsure why references to me (Doolittle) include my initials or middle name (Ford) while others just have the last name only. Ford is not part of my surname.

Introduction. I think it is (or should be) more correct to say that ENCODE critics like me don't so much insist that everyone use the SE definition of function, as that they be more precise about the definition they ARE using. Many genomicists and molecular biologist seem to think the "function" is unproblematic. This might be so if all activities arise by selection, and I think that most genomicists and molecular biologists are indeed pan-adaptationists, although not realizing it! Authors do say that something like this in the second paragraph on this page.

Introduction. I like the idea of thinking about the "function" of newly arising genes. I think I agree with them that the idea of a gene implies a function, but wonder whether one would say, of a newly arisen stretch of DNA with a promoter, open reading frame producing a substantially long protein with no activity whatever, and terminator, that it was not a gene? One could easily make such a thing in the lab. Would we not call it a "gene", or would we also require that it has a biological function?

Authors do problematize something like this, though having a de novo gene that causes cancer seems to be over-problematizing. I think many who would say "Gene X functions in the development of cancer [or even some disease that does not itself evolve]" would balk at saying that "The function of gene X is to cause cancer".

Not altogether sure what they mean "hinder scientific development". They say (and I agree) that "function" and "matter" (as in "Does it matter?") are often conflated, but possibly any further definition of function will do.

Subsection “A model of function for de novogene birth research”. Pittsburgh model seems very useful and appropriate. Was it just invented for this paper, or does it have a provenance? I think the former, but this should be made clear.

Subsection “The multiple meanings of function hinder scientific development in the field of de novogene birth”. Only 20 abstracts? Seems a small number, and it is unclear to me why they did not do more. Should be made very clear that all these 20 describe de novo originating genes. And actually, why not abstracts of paper in genomics/molecular biology more generally. Does the Pittsburgh model not apply equally to all?

Subsection “The multiple meanings of function hinder scientific development in the field of de novogene birth. Not sure what "consensus" means. Did all authors get together and try to agree (which is what I think the term usually means) or did they just take the majority of independent opinions.

Subsection “Interpretation and recommendations”. Totally agree that when the word "function" is used, it should be qualified, or else it should not be used.

Overall, a nice paper and I very much like the "Pittsburgh model". As a suggestion for further work, this is fine. It could so easily have been applied to a much larger number of readers (think of grad students in different biological disciplines, for instance) that I wonder why it wasn't. Or indeed to abstracts in genomics/molecular biology more generally.

Reviewer #2:

The paper seeks to establish a hierarchy of clarifications for the word "function", to help reduce confusion. The authors propose a 5-level classification of Expression (E), Capacities (C), Interactions (I), Physiological Implications (PI), and Evolutionary Implications (EI), and then seek to classify the use of the word "function" in 20 papers into one of the 5 classes, or "vague", if it is unclear which use is meant. The four authors independently assign one of the 6 labels to each instance, and (to their surprise, it seems) find that even the authors themselves cannot agree 88% of the time which of the classes should be used. The authors conclude that this disagreement demonstrates how important their classification scheme is, but I would argue it also demonstrates that such a scheme is very unlikely to catch on.

Specific comments:

1) Table 1: The classification scheme is problematic. For example, one could argue that E, C, and I could be referred to as "biochemical activity", not "function", despite the unfortunate naming of the field of "Functional Genomics" (which mostly concerns itself with biochemical activity). This 'activity-vs-function' distinction is consistent with the much less frequent usage in Figure 1C of these three categories compared to the last two.

2) Table 1: Still on the classification scheme, one could also argue, the top contenders for the word "function", PI and EI are already quite extensively distinguished in the literature in the names of Selected Effect (SE) and Causal Roles (CR). SE seems to be the same as EI, in which case a new term may not be needed. CR seems to be most closely aligned with PI.

3) Introduction: The paper frames itself in the context of ENCODE, and then shifts gears and focuses on the field of gene birth. The authors should take a broader perspective of the utilization of the word function in their Introduction, or take a much narrower perspective focusing on gene birth. The current Introduction distracts instead, and feels more like a bait-and-switch. If the authors choose to start with ENCODE in the Introduction, they should instead take on the task of classifying the 300-some papers by the ENCODE consortium instead.

4) Figure 1: The classification task undertaken is quite modest. Given the ambitious stated goal of the paper, a much larger number of papers should be utilized, to avoid the biases that can come from small datasets.

5) Figure 1: The authors should also provide the full classification by each author of each instance, to enable analysis of which terms are most confused, which are most ambiguous, which specific pairs are switched, etc. The authors should also carry out some of that analysis as well, in a larger sample.

6) Figure 1: The authors should engage third-party scientists (perhaps student volunteers from a graduate program or students from a class) to read the instructions and then classify the words, in order to get a larger sample.

7) Introduction: The first sentence of the third paragraph suggests a very narrow view of the debate surrounding the ENCODE project and the definition of function therein. This is probably not an area that the authors want to get into. In particular, searching for "ENCODE debacle" in Google Scholar returns only one paper, which has never been cited. Searching in Google returns only blog posts by Dan Graur and other angry bloggers, which is probably not the view that the authors want to align themselves with. A closer reading of the original ENCODE 2012 paper provides upfront a definition of "biochemical function" (which perhaps should be referred to as "biochemical activity", despite again the unfortunate naming of "functional assay" and "functional genomics"). It then uses that definition, and very clearly indicates that only a small fraction of the genome is under evolutionary selection. Even the criticisms of ENCODE have primarily cited the press articles written by news authors, not scientists, that claim that 80% of the genome is functional. Anyways, I would skip that whole section if the authors don't want to re-open a very large can worms.

8) Introduction: The selected effect that gave rise to a trait or a genomic region may be quite different from the current functional roles of that trait or genomic region. Thus, equating function to "selected effect" may be inaccurate as well.

9) Introduction: "scientists cannot agree on the number of functional genes in the human genome" is a peculiar statement to accompany the Pertea and Jungreis papers. Briefly, Pertea claims to discover thousands of new genes, Jungreis claims that Pertea made specific mistakes resulting in exclusively false positives. Yes, there is debate, but the Pertea paper is not a reference for this statement. This sentence alone indicates a lot of nonchalance on the part of the authors about dismissing the state of broad fields that they should be much more cautious about, especially in a paper that seeks to bring rigor to the field.

10) Introduction, "practically zero". This is again a great oversimplification of a rich field of gene birth, and does not reflect well on how scholarly the authors should be.

11) Introduction: "gene". Why choose this word, rather than "functional". The word "gene" itself has a long history of differing definitions and great debate. Once more, throwing this word around without much thought seems out of place for a paper that seeks to be scholarly.

12) Subsection “A model of function for de novo gene birth research”. The example seems contrived. Why conflate the concept of de novo gene birth with the fact that many disease relevant mutations are not in selected elements. These should be two separate examples. Ex1: mutations that lead to disease but lie in non-conserved regions. Ex2: gene birth example.

13) Subsection “A model of function for de novo gene birth research”: Please reword to avoid the word "fantastic".

14) Subsection “A model of function for de novo gene birth research”: "Perturbation effect" should feature within this list. Perturbation by experimental intervention is one example. Perturbation by natural genetic variation is another. Perhaps both should be separate entries on their classification.

15) Subsection “A model of function for de novo gene birth research”: Naming this Pittsburgh after the affiliation of the last author seems inappropriate. Does every scientist in Pittsburgh agree? Why not "Carvunis", since she's the only author from Pittsburgh? Why not "our" model, and let others name it "the Carvunis model".

16) Table 1: Should this be a hierarchy? Why not a list of checkboxes/attributes. Different papers show evidence of one without evidence of the other.

17) Table 1: Order between C and E is unclear. Perhaps capacities precede expression.

18) Table 1: E: does mere presence of DNA make every DNA segment functional?

19) Table 1: Selfish elements defy this classification. Distinct SE and CR 'functions'. Similarly, "runaway selection" for traits that are detrimental to biological functions challenges this definition. Both should be discussed.

20) Page 7, second paragraph: Why not address "surplus meaning"? Please expand how it could be addressed. Roux 2014 also brings up many additional important points worth discussing at greater length.

21)Subsection “The multiple meanings of function hinder scientific development in the field of de novo gene birth”,"only 12% were unanimous":

- need heatmap of misclassifications, table of all data.

- which ones were most confused and why

- this result perhaps indicates that nomenclature would not have helped, may be either too ambiguous or overspecified

- an independent cohort of students / trainees / colleagues would be useful in evaluating approach

- more than 20 papers would greatly help.

22) Subsection “The multiple meanings of function hinder scientific development in the field of de novo gene birth”. "again supporting our hypothesis...". Perhaps also suggesting that the proposed classification scheme does not work?

23) Subsection “Interpretation and recommendations”: for "older" genes the SE and CR functions are more likely to differ, and thus focusing on gene birth may bias authors' perspective on one hand, and may also lead to non-representative results on the other hand.

*Reviewer #3:*

The authors address an interesting question regarding the philosophical issue related to the meaning(s) the word "function", and in particular the question whether a selective process is required to legitimate the usage of the word "function". Though extensive philosophical debate has occurred within the last decades, little attention has been paid to the usage of the word by the biologist themselves. The birth event of de novo gene, understood as the event that occurs prior to any selective process, is an interesting limit case to see whether biologists use or not the word "function" to characterize some particular properties of these genes. However, as it is presented in the manuscript, the work exhibits several conceptual and methodological weaknesses.

A) The philosophical issue of de F-word:

1) As stated by the authors, the philosophical debate regarding the function is about (a) what a function means and (b) how this meaning is legitimated. The (a) debate is between, as formulated by Cummins the "How does it work?" question (causal role theories) and the "Why is it here?" question (teleological-etiological theories). In the first case, legitimation is grounded on the functional (or systemic) analysis of the biological system studied, in the later one, it is grounded, for the selected effect theories, on the existence of a past selective process. Hence, in these theories, the reference to evolutionary implications is not that the effect of a trait is adaptive, but that its presence is the consequence of a past process of adaptation. What legitimates a function is the past, not the current, fitness of a trait, because current fitness is not a legitimate response at the question "why an entity does what it does". Regarding this question, the de novo gene occurrence, seen as a transition between no-function to function, is an interesting case. Indeed, the de novo occurrence of a trait cannot be explained by a selective effect, even if it has physiological implications, and even if it considered as adaptive (which means that it has "positive" physiological implications). But his critical point is not taken into account by the authors in their study, including in their definition of "evolutionary implications".

2) Regarding the teleological-etiological theories, in parallel with the selected-effect theories of function, a new approach has emerged in the last decade, the so-called organizational approach of function (OAP). For these theories, what a function means is, as for the selected effect theories, "Why is it here?", but the legitimation is not grounded on a selective process but on organizational properties, basically the existence of a causal loop between the causal role of a trait in the self-maintenance of the system, and the production and maintenance of a trait by the system. The application of this theory to the case of de novo gene birth is interesting, but there is no reference to this theory of function in the manuscript.

3) Taking into account the philosophical issue of the F-word, the emergence of de novo genes is problematic only for the supporters of the selected-effect theories; it is not for the advocates of the causal role theory and the OAP. Indeed, the main critic of the OAP against the selected effect theory is precisely that this theory cannot account for the emergence of function. The title is hence rather misleading, since it refers to a privileged theory of function. The word "evolution" is, in the same idea, also confusing. Why use it rather than emergence? Indeed, if the de novo emergence of gene "function" has anything specific regarding the F-word philosophical issue, it is that it is de novo, i.e., it has not (or not yet) been submitted to an past selective pressure. If the existence of de novo genes and their effects can be interpreted as a consequence of a selective pressure, they are no longer "de novo". The meaning of "evolution", used for de novo gene, should be precise. If it precisely means the birth event of a new gene, this event occurs prior to any selection, because selection can act only by differential screening between already existing traits. In this meaning, "emergence" seems more adequate.

B) Semantic plurality of the F-word and conceptual confusion.

4) The authors claim that the existence of several meanings of "function" is a source of confusion between biologists, but there is no reported evidence of such a confusion. Actually, the word "function" is used in biological sciences at least from the sixteenth century, without particular debates within the biologist about what this word means. The fact that there is a philosophical debate about the legitimacy of the meaning of a notion like "function", and the fact that this notion is a useful or a confusing conceptual tool is 2 different questions. The authors should precise if the object of their work is to contribute to the philosophical debate by clarifying the meaning of the concept by the biologists, or to highlight to existence of real confusion or discrepancies in the usage of the word by the biologists, specifically in the field of de novo gene research.

5) In their rhetorical analysis, the authors have considered that the meaning of "function" is similar when it is used as a name ("assessment of function"), a verb (MDF1 functions in two important pathways"), and adjective ("functional"). However, some philosophers have pointed out that "a function", "to function", and "functional" have different meanings, expressed by different words in other languages than English. For example, in the cited though experiment, scientist who may be reluctant to say that "the function of the pro-oncogenic gene is to generate cancer", but agree to say that "the function of the anti-oncogenic gene is to prevent cancer", would agree to say that both genes are "functional". The authors should provide the numbers of instances of the different words used and justify that there is no different meanings statistically associated to the name, the verb and the adjective.

6) The existence of different meanings of a notion does not necessarily imply that the notion is conceptually confusing, if the different meanings are complementary and not contradictory. Actually, the fact that the authors have been able to attribute, according to the "Pittsburg model of function", identified meanings to the usage of the term "function" can be interpreted as a proof that the word is not so confusing. Additionally, the presentation of the "Pittsburg model of function" in table 1 show causal relationships between the different meanings, symbolized by arrows from the lowest line (expression) to the highest (evolutionary implication). It seems to me that these arrows are a "surplus meaning" added by the authors from their rhetorical analyses, rather than relationships identified by the rhetorical analysis itself, but, anyway, as it is presented, the "Pittsburg model of function" is nothing more than the classical bottom-up view of causal relationships "from genes to function".

7) I do not understand on which argument the authors can say that the data presented in Figure 1 support the idea that the literature is hard to interpret. As said previously, if the different meanings are causally subordinated, the interpretation of the data does not seem so difficult, including in the fact that different meanings can be embedded in the term "function". Can the authors provide examples of misinterpretations?

8) Regarding the usage of the word "function", the main conclusion of the rhetorical analysis is that the large majority of the meanings are physiological implications and evolutionary implications, which represent almost all of the instances (40 on 42). The authors should discuss the implications of such a rather "integrative" or "high level" meaning of function on the philosophical debate about what a function mean. The fact that, as stated by the authors, "evidently, neither technique gives direct insights into evolutionary implications" should also be discussed. Do the authors consider that this methodological difference impacts the legitimation of the meaning of function?

C) Methods

9) Methodologically, the fact that the literature has been studied only on abstracts, and not on the full papers, is a serious limitation of the work. Why the authors haven't work on the full papers? In particular, the "vague" classification on abstracts may be not so vague in the full paper. The number of abstracts is also small, as is the number of coders.

10) Another problem is how the abstracts have been selected. A sample can be considered as representative only if it had been randomly picked up, which is not the case in the study. At the best, it can be considered as illustrative. Is the fact that the person who selected the abstracts is an "expert in the field" supposed to be, by itself, a guarantee of objectivity?

11) Another important methodological limitation is the fact that the sample of abstracts has been used to build the model, to validate it, and to use it is basically the same; indeed, 17 of the 20 abstracts where used for model refinement, and the remaining 3 for its validation. After that, the same 20 abstracts where used for quantitative analysis and "hypothesis testing". I do not clearly see what are the tested hypotheses? But the fact that the same set of material and the same set of coders where used all along the process it for me a serious methodological limitation.

12) Taking together, these methodological considerations greatly impact the accuracy of the study. Independently from these methodological limitations, the question arises of how such a methodology brings novel insights is the F-word issue, and, in particular, the real value of the so-called "Pittsburg model of function". As said before, its presentation in table 1 seems no more than the classical bottom-up "layer cake" view of stratified levels of organization, from molecules to organisms interacting with their environmental conditions (which is, by the way, seriously challenged by several philosophical studies). The authors should hence precise what is new in the so-called "model", either philosophically or conceptually.

Reviewer #4:

General assessment: this manuscript is an important and thoroughly argued contribution to *eLife*. It is particularly notable how accessibly the manuscript is written for a multi-disciplinary audience. The arguments are timely, and the need for this work is justified in the opening section. The description of the findings and later description of the approach are detailed and explanatory, with interpretation provided as relevant. I highly recommend this piece for publication.

[Editors' note: further revisions were requested prior to acceptance, as described below.]

Thank you for submitting the revised version of "The meanings of 'function' in biology and the problematic case of de novo gene emergence". This version has been seen by two of the reviewers who reviewed the original version (Ford Doolittle – Reviewer #1; Etienne Roux – Reviewer #3), and their comments are below. I would like to invite you to submit a second revised version that addresses these comments, and also the additional comments listed below them.

Reviewer #1:

I still wish that the authors had looked at more than 20 abstracts, but the writing is very clear and the intellectual clarification this promises is enough, in my view, to justify publication. And I'm sure that many in future will refer (if sometimes critically) to the "Pittsburgh Model of Function". So, I think this can be published as is, though I have these residual quibbles.

1) They say, of the top level in that model, that.…"Evolutionary Implications can refer to selection upon a trait driven by the object in the past, as in selected effect (Millikan, 1989), but it can equally describe novel adaptive effects, as one may expect in gene birth." Not sure what an "adaptive effect" is. If selection is a force and the effect is fitness-enhancing in environment E, then from the very instant that organism O finds itself in E, this "adaptive effect" is under selection, though possibly only purifying selection if it was previously fixed in another environment, where it had no, or another, effect on fitness. If selection is an outcome, then nothing is a selected effect unless actual mutations have actually been eliminated by purifying selection just yesterday, which seems silly. Any species emerging from a bottleneck has no functions, by that account.

2) The statement "For a genomic sequence to be labelled as a gene, it must by definition have a function." needs some further elaboration or justification (as in the response to reviewers.

3) And to say "The molecular objects of study are thus transitioning between a state without a function and a state with a function. They cannot have a function by the selected effect definition, since their existence cannot be explained by a past selection." seems a little problematic. If by a "new gene" one means a gene that was never detected in any other species, but it is fixed in the species in which it is detected, and that species has a reasonable population size, then it likely was selected for, in that species, even if we do not know why. And I suspect that most of the literature on de novo genes describes such situations, not situations in which the "new gene" is a rare polymorphism in the species in which it is described. So it's really only the parent of the new gene that is not a gene: at the very instant that a replicate of this non-gene parent starts to increase in frequency in its population because of a fitness advantage to its bearers, it has acquired 'function' and becomes a bona fide gene, I think. I don't see this as a gradual process or transition, which the authors in their next paragraph, seem to imply. Of course this could lead further into a discussion of whether selection is a force or an outcome or whatever, and what does fitness mean anyway, but I don't think authors need to go there to be clearer on this point.

Reviewer #3:

The authors have greatly improved the quality of the manuscript and correctly respond to the majority of my comments. There is however 2 points for which the response given by the authors let me unsatisfied.

1) The first one is about the hierarchical relationships introduced by the authors in their model, and symbolized by ascendant arrows in Table 1. Since, as acknowledged by the authors, these relationships are a "surplus" meaning injected in, and not extracted from, the rhetorical analysis (so the model is not purely data-driven), the question remains of where this surplus meaning comes from, and why is it needed. As I have said in my initial comment, this view of hierarchically organized relationships is not new, and corresponds to the classical bottom-up causal relationships from lower (gene) to higher level of organization. This is not a question of logic, but a question of causal relationship between different levels of properties, similar to the classical causal relationship usually expressed by arrows from gene to proteins to pathways to subcellular mechanisms to cells to tissues to organs, etc. (see for example Noble, 2006). Saying so, I do not mean that the model is irrelevant, but that, since the authors privilege a bottom-up view of causal relationships between properties expressed at different level of organization which is not consensual, they should be more explicit about the import of this causal pattern in their model (including bibliographical references) and its justification. Also, some expressions seem misleading or inappropriate. For example, I don't see what is a "logical flow of genetic information" from lower to upper levels of organizational properties, and top-down causal relationships also exist (typically, gene expression is a consequence of interactions). The notion of "spatiotemporal relationship between concepts" does not seem to me relevant. First, the spatiotemporal relationships to which the authors refer are between properties of biological entities, not between concepts, and, in the case of de novo gene, the spatiotemporal relationship between interactions and expression is primarily from interaction to expression, not the opposite, since the expression of a de novo gene cannot occur without prior interaction of the DNA segment with a lot of different proteins involved in a complex interaction network.

If we admit the relevance of the directional necessary-but-not sufficient causal relationship between from Expression to Evolutionary implications, it can be logically inferred that each level of meaning of function is inclusive of all the meanings of lower levels. The authors should be more explicit in which way this modifies a model of function, compared to a model without these hierarchical relationships (basically, they should explain why they have chosen to add this hierarchical relationship and how it improves the model). Since previous studies have already identified different notions of functions (see for example Wouters, 2003), the authors should refer to these previous studies in their discussion.

2) The second point is about the claim, which is also a section of the article, that "The multiple meanings of function hinder scientific communication in the field of de novo gene birth". As I have previously explained, I don't think that, from a biologist viewpoint, the notion of function is confusing, and that is impairs the scientific communication. However, the point is not that I disagree with the authors, but is that the study of the authors provides no evidence for such a claim. The only factual evidence is that, in a set of four encoders, among whom one is an expert is the field, one non-consensual case remains on the 20 abstracts analyzed. Due to the numerous limitations of the study, both in its conception and methodology, these results are clearly unsubstantial; also, non-consensus and discrepancies between members can be interpreted as the inability of the so-called Pittsburg model to account for the actual use of the concept by the researchers themselves. Due to all these limitations, I find such a claim, and the final recommendations, an overinterpretation of the results and, in my opinion, should be removed, or at least reformulated in a less affirmative way.

---

## [Author Response]

Thank you very much for your comments. We are thrilled to see in many of your compliments and encouragements that the importance of rhetorical analysis in scientific communication is recognized as a means to further the conversation about function for an interdisciplinary audience, and we are grateful for the constructive criticisms that you provided. We find our revised manuscript much improved thanks to your suggestions.

Among the most helpful were similar points raised independently by several reviewers. In our response letter, we start by addressing these common points in a dedicated section where they are grouped by topic: #1: Scope, #2: Table 1, and #3: Cancer example and usage of function.

We then address the remaining comments one by one in the next section.

1) Scope:

Several reviewers pointed out the limited scope of our study. These limitations include 1) that we only analyzed a small number of abstracts and 2) that we focused on the de novogene birth field. The specific reviewer comments are listed below and our response follows.

Reviewer 1:Subsection “The multiple meanings of function hinder scientific development in the field of de novo gene birth”. Only 20 abstracts? Seems a small number, and it is unclear to me why they did not do more. Should be made very clear that all these 20 describe de novo originating genes. And actually, why not abstracts of paper in genomics/molecular biology more generally. Does the Pittsburgh model not apply equally to all?Overall, a nice paper and I very much like the "Pittsburgh model". As a suggestion for further work, this is fine. It could so easily have been applied to a much larger number of readers (think of grad students in different biological disciplines, for instance) that I wonder why it wasn't. Or indeed to abstracts in genomics/molecular biology more generally.Reviewer 2:4) Figure 1: The classification task undertaken is quite modest. Given the ambitious stated goal of the paper, a much larger number of papers should be utilized, to avoid the biases that can come from small datasets.5) Figure 1: The authors should also provide the full classification by each author of each instance, to enable analysis of which terms are most confused, which are most ambiguous, which specific pairs are switched, etc. The authors should also carry out some of that analysis as well, in a larger sample.6) Figure 1: The authors should engage third-party scientists (perhaps student volunteers from a graduate program or students from a class) to read the instructions and then classify the words, in order to get a larger sample.19) Table 1: Selfish elements defy this classification. Distinct SE and CR 'functions'. Similarly, "runaway selection" for traits that are detrimental to biological functions challenges this definition. Both should be discussed.21) Subsection “The multiple meanings of function hinder scientific development in the field of de novo gene birth”, "only 12% were unanimous":- need heatmap of misclassifications, table of all data.[…]- an independent cohort of students / trainees / colleagues would be useful in evaluating approach- more than 20 papers would greatly help.23) Subsection “Interpretation and recommendations”: second paragraph, first sentence: for "older" genes the SE and CR functions are more likely to differ, and thus focusing on gene birth may bias authors' perspective on one hand, and may also lead to non-representative results on the other hand.Reviewer 3:9) Methodologically, the fact that the literature has been studied only on abstracts, and not on the full papers, is a serious limitation of the work. Why the authors haven't work on the full papers? In particular, the "vague" classification on abstracts may be not so vague in the full paper. The number of abstracts is also small, as is the number of coders.

We wholeheartedly agree that 20 abstracts is a very small sample. In fact, we clearly outlined this a major limitation of our work in the submitted manuscript:

“The Pittsburgh model has important limitations. Tested on only 20 abstracts within a single field of study, by no means does it represent a complete ontology of function.”

In the revised manuscript, we further clarify this limitation to make clear that the abstracts are solely within the field of de novo gene birth:

“The Pittsburgh model has important limitations. Tested on only 20 abstracts related to the single field of de novo gene birth, by no means does it represent a complete ontology of function.”

And by enouncing the number of abstracts earlier and the nature of our methodology earlier in the text:

**“**In order to derive an initial model of function adapted to de novo gene birth research, we first rhetorically analyzed the scientific literature in the field together with philosophical publications about genomic function. We then applied the constant comparative method of the grounded theory of social sciences (Glaser and Strauss, 1967) to samples of 20 published abstracts in the field (see Approach). Through an organized, iterative process of defining and discussing usages of the term, we inductively converged on the interpretation that, in this set of abstracts, authors writing about the function of a molecular object were almost always describing one or more of the following properties of the object: Expression, Capacities, Interactions, Physiological Implications and Evolutionary Implications. These properties represent five meanings of function that are defined in Table 1.”

We agree that the small number of abstracts contrasts with our ambitious stated goal as noted by reviewer 2. We have thus toned down our goal in the revised manuscript:

“As an interdisciplinary group of scholars, we sought to understand to what extent the existence of differing meanings of “function” actually impairs the scientific enterprise.”

That being said, we would like to point out that it is well established in the field of rhetoric that a study can never capture the entire discourse, and therefore small text samples remain useful as long as they enable scholars to construct a specific argument. This is specified in our revised manuscript to justify the value of our work, although we fully agree that more abstracts would further strengthen the study:

“The papers were chosen to span a variety of journals, countries, citation counts, model organisms, methodologies and scope, in order to derive a context specific rhetorical argument (McGee, 1990)”.

It is possible that analysis of the full papers may inform upon the authors view point and possibly assist in the interpretation of “vague” assignments of function. It is also possible that with more coders, some could discern a meaning we did not. However, given that our study focuses on language, our coding rules were to decipher the meaning from the written sentences, rather than from the actual results of the paper. This was explained in the Approach section of our submitted manuscript:

“assignments must reflect what the coder thinks that the author meant given the context of the sentence and abstract, rather than what the coder thinks the scientific data presented actually demonstrates assignments must preferentially derive from the sentence in which the term is used alone”.

In our revised manuscript, we have further clarified this stipulation to justify our use of abstracts:

“Each member of our team independently assigned one or more meaning to each instance of the word function found in our database of abstracts, gathering contextual evidence primarily from the sentence in which the term was used”.

Finally, we find it unlikely that reading the entire paper would have given textual evidence on words used in the abstracts in sentences such as:

“Orphans are an enigmatic portion of the genome since their origin and function are mostly unknown and they typically make up 10 to 30% of all genes in a genome." (Wissler et al., 2013)

Our entire data set, including all independent and consensus assignments for each of the 42 instances, is included in our revised manuscript as Figure 1—source data 1. This will be useful to readers interested in the vague assignments or interested in bettering our model.

We deliberately focused our research in the field of de novo gene birth because it represents an interesting edge case, and it is highly interdisciplinary and therefore rich in diverse technologies and perspectives. We make no claim that it applies to any other type of genetic elements such as selfish elements. We have clarified this in the revised manuscript:

“The Pittsburgh model has important limitations. Tested on only 20 abstracts related to the single field of de novo gene birth, by no means does it represent a complete ontology of function. Constructed in a field-specific manner, it may or may not extend to genomic elements outside of the gene birth framework, such as selfish elements or promoters. It cannot be readily applied to biological objects other than DNA, RNA and protein objects.”

Like reviewer 1, we suspect that our model might be suitable more broadly to genomics/molecular biology research, but this is not a hypothesis we have tested so we do not make that claim. We have revised our discussion to propose it explicitly as a suggestion of future work:

“We encourage researchers to refine the model, adapt it to other subfields of genomics and other types of biological objects (metabolites, cell types, organs), and propose alternatives. One can envision, for instance, exploiting the power of computational natural language processing to generate field-specific models by automatic analysis of large bodies of literature (Friedman et al., 2001; Groth et al., 2016), perhaps using our modest manual analysis as a training set.”

We respectfully advance that focusing on one field can be considered a strength rather than a limitation. Indeed, as we and others have pointed out, general philosophical theories of function (such as CR) are not adopted in scientific writing practice unless they are tailored to the problems scientists are dealing with. These problems are field-specific. Our model is not general and does not claim to be. It cannot be applied to all types of biological objects or questions, yet we expect it to become useful in the scientific area it was customized for. In our revised manuscript, we highlight that the concept of function deserves field-specific attention:

“Function is a concept that depends on the methodological practices, measurement procedures and habits of scientists. As these change, so too will the concept of function change and adapt to specific subfields of research, because it is contingent and always in a state of flux.”

This manuscript represents the first rhetorical analysis that qualifies how function is used in scientific writing. We feel that the novelty of our interdisciplinary approach and the practical implications of our argument somewhat mitigates the limited size of our sample of texts. We hope that it will inspire more work to improve and extend our model (or build better ones inspired by our approach!) to broader fields.

2) Table 1:

Reviewers 2 and 3 raised important questions about our model presented in Table 1. The specific reviewer comments are listed below, followed by our response and then by the list of changes we made in the revised manuscript to address these questions. We are particularly grateful for this set of comments which have led to a substantial improvement of our manuscript.

Reviewer 2:1) Table 1: The classification scheme is problematic. For example, one could argue that E, C, and I could be referred to as "biochemical activity", not "function", despite the unfortunate naming of the field of "Functional Genomics" (which mostly concerns itself with biochemical activity). This 'activity-vs-function' distinction is consistent with the much less frequent usage in Figure 1C of these three categories compared to the last two.2) Table 1: Still on the classification scheme, one could also argue, the top contenders for the word "function", PI and EI are already quite extensively distinguished in the literature in the names of Selected Effect (SE) and Causal Roles (CR). SE seems to be the same as EI, in which case a new term may not be needed. CR seems to be most closely aligned with PI.14) Subsection “A model of function for de novo gene birth research”: "Perturbation effect" should feature within this list. Perturbation by experimental intervention is one example. Perturbation by natural genetic variation is another. Perhaps both should be separate entries on their classification.16) Table 1: Should this be a hierarchy? Why not a list of checkboxes/attributes. Different papers show evidence of one without evidence of the other.17) Table 1: Order between C and E is unclear. Perhaps capacities precede expression.19) Table 1: Selfish elements defy this classification. Distinct SE and CR 'functions'. Similarly, "runaway selection" for traits that are detrimental to biological functions challenges this definition. Both should be discussed.20) Subsection “A model of function for de novo gene birth research”: Why not address "surplus meaning"? Please expand how it could be addressed.[…]Reviewer 3:1) As stated by the authors, the philosophical debate regarding the function is about (a) what a function means and (b) how this meaning is legitimated. The (a) debate is between, as formulated by Cummins the "How does it work?" question (causal role theories) and the "Why is it here?" question (teleological-etiological theories). In the first case, legitimation is grounded on the functional (or systemic) analysis of the biological system studied, in the later one, it is grounded, for the selected effect theories, on the existence of a past selective process. Hence, in these theories, the reference to evolutionary implications is not that the effect of a trait is adaptive, but that its presence is the consequence of a past process of adaptation. What legitimates a function is the past, not the current, fitness of a trait, because current fitness is not a legitimate response at the question "why an entity does what it does". Regarding this question, the de novo gene occurrence, seen as a transition between no-function to function, is an interesting case. Indeed, the de novo occurrence of a trait cannot be explained by a selective effect, even if it has physiological implications, and even if it considered as adaptive (which means that it has "positive" physiological implications). But his critical point is not taken into account by the authors in their study, including in their definition of "evolutionary implications".6) The existence of different meanings of a notion does not necessarily imply that the notion is conceptually confusing, if the different meanings are complementary and not contradictory. Actually, the fact that the authors have been able to attribute, according to the "Pittsburg model of function", identified meanings to the usage of the term "function" can be interpreted as a proof that the word is not so confusing. Additionally, the presentation of the "Pittsburg model of function" in table 1 show causal relationships between the different meanings, symbolized by arrows from the lowest line (expression) to the highest (evolutionary implication). It seems to me that these arrows are a "surplus meaning" added by the authors from their rhetorical analyses, rather than relationships identified by the rhetorical analysis itself, but, anyway, as it is presented, the "Pittsburg model of function" is nothing more than the classical bottom-up view of causal relationships "from genes to function".7) I do not understand on which argument the authors can say that the data presented in Figure 1 support the idea that the literature is hard to interpret. As said previously, if the different meanings are causally subordinated, the interpretation of the data does not seem so difficult, including in the fact that different meanings can be embedded in the term "function". Can the authors provide examples of misinterpretations?12) Taking together, these methodological considerations greatly impact the accuracy of the study. Independently from these methodological limitations, the question arises of how such a methodology brings novel insights is the F-word issue, and, in particular, the real value of the so-called "Pittsburg model of function". As said before, its presentation in table 1 seems no more than the classical bottom-up "layer cake" view of stratified levels of organization, from molecules to organisms interacting with their environmental conditions (which is, by the way, seriously challenged by several philosophical studies). The authors should hence precise what is new in the so-called "model", either philosophically or conceptually.

First, we address reviewer 2’s suggestion that Expression, Capacities and Interactions could be referred to as biochemical activity.

We agree that Expression and, in many cases, Interactions can be considered biochemical activities. However, in the context of using our model as a lexicon, we feel it will be more useful to use these terms rather than biochemical activity in writing. To keep with the examples, we cite in our manuscript, the results of an RNA seq experiment are more precisely described as informing upon expression, and those of a yeast two-hybrid experiment as informing upon interactions, than using the same term of biochemical activity or effect for both. We respectfully do not understand how Capacities, defined in Table 1 as the intrinsic physical properties of the object under investigation (such as a predicted protein’s conformational dynamics), may be considered biochemical activities (which we understand as requiring a system where the object can be active). We end by noting that reviewer 1, who proposed the effect/function distinction, did not remark on similarities with our model and stated that he “very much like[s] the Pittsburgh model”.

Second, we clarify the relationship between our model and the Causal Role / Selected Effect paradigm (CR/SE) and explain why Perturbation Effect would not fit within our framework.

We would like to clarify that our model, unlike CR/SE, does not aim to define proper use of the word function based on philosophical arguments. We developed it as a tool to help understand meanings that are already in circulation in the field. The meanings we identified are associated with the measurement context, rather than the theoretical context of function (Laubichler et al., 2015). These five meanings correspond to natural properties of the object in time and space. As such, we feel that perturbation effects would not fit in our model. Indeed, perturbation effects by experimental intervention can be seen a means to explore the natural properties of the object, rather than the properties themselves. And perturbation by natural variation are implicitly included as one of the many types of Evolutionary Implications that can be described with our model.

Furthermore, we advance that there is not a direct one-to-one mapping between our 5 meanings and the CR/SE theories of function. A Causal Role function could be inferred from any of Expression, Interactions and Physiological Implications. Physiological Implications, however, also map to Selected Effect, when they describe the reason why an object is present from a selection perspective. Selected Effect also maps indeed to Evolutionary Implications, in the sense that the reason why an object was selected in the past can also be viewed from the perspective of how much it has influenced population dynamics. However, Evolutionary Implications is very broad, defined as the objects influence on population dynamics. Compared to Selected Effect, it is both more precise (it is only concerned with effects across generations, not the nature of underlying physiological mechanism) and more general. It can describe many sorts of effects across generations, including the example of runaway selection mentioned by reviewer 2, an effect from the past as in Selected Effect, or a novel adaptive effect as in the context of gene birth as mentioned by reviewer 3. Our model explicitly describes some of the properties of the object that could be assigned to both CR and SE functions. Furthermore, as we commented in response to reviewer 1, an interesting aspect of our model is that a natural or artificial sequence that does not have any activity can still be described in terms of its Expression and Capacities, although this is not discussed explicitly in our manuscript.

Third, we discuss reviewer 3’s suggestion that our model is akin to “the classical bottom-up layer cake” “from gene to function”, “stratified levels of organization, from molecules to organisms interacting with their environmental conditions”, and how this notion relates to the order of the rows in Table 1 questioned by reviewer 2.

We are not aware that a model including our 5 meanings in the order we proposed has ever been proposed previously. We inductively constructed it from careful reading of literature in this area and analysis of our abstract collection. As we expose below, we now see that the arrows we used in the model could have led to the assumption that the five meanings were casually connected and this, in turn, might explain why parallels between those models and ours were pointed out. That said, we readily agree that our model fits in with current understandings of genomics, genetics, systems biology and evolutionary biology – this was a feature we were striving for, and which we pointed out in our submitted manuscript:

“Conveniently, these five meanings map to a general interpretation of the logical flow of genetic information over time and space.”

We were careful in using “a” general interpretation in this sentence, because there are many other ways current understandings of molecular biology can be interpreted. One other possible interpretation was suggested by reviewer 2:

“*Perhaps capacities precede expression.**”***

We hedged the universality of our model in our submitted manuscript:

“by no means does it represent a complete ontology of function.”

Finally, we address the remaining problems raised by the two reviewers, which relate to the hierarchical organization of the 5 meanings indicated by vertical arrows.

We had regrettably omitted to define these arrows in the legend of Table 1 in our submitted manuscript. Thanks to these reviewer’s comments, this important omission has been corrected in the revised manuscript. The arrows were not intended to represent strict causal relationships, but instead that a given meaning is necessary but not sufficient for inferring meanings above it in the hierarchy. For example, an interaction does not necessarily imply a physiological implication, but a physiological implication does imply the existence of at least one interaction. This was explained in the coding rules section of our submitted manuscript:

“when a single instance of the term is interpreted as multiple meanings, and context does not help distinguish them, then the meaning that is furthest in the progression from Expression to Evolutionary Implications must be coded (Table 1)”.

And exemplified in the main text:

“so too, establishing that a locus is expressed does not imply that the corresponding product necessarily interacts with other elements in the cell.”

We hope that, having clarified the meaning of the arrows in the revised manuscript, it is now clear why our model is a hierarchy, and why Figure 1 supports that the literature is hard to interpret (as it effectively shows that the word function can be interpreted differently by readers within the same sentence). We agree with reviewer 3 that these arrows are indeed a “surplus meaning” rather than the result of the textual analysis itself. We have added this in our revised manuscript, leaving out the expression “surplus meaning” itself so that the reader does not get confused between the surplus meaning of our arrows, and the surplus meaning of teleology. We are grateful for the opportunity to correct our omission in the revised manuscript, to avoid misunderstandings such as those raised in reviewer 2 point 16 or reviewer 3 points 6 and 7.

Again, we are very grateful for these very useful comments and suggestions which provide us the opportunity to further clarify how our model differs from existing ones, and what the arrows mean. Our revised manuscript includes the following changes:

- Specifying what is new in our model relative to activity/function and CR/SE:

“This model presents practical, field-specific advantages relative to pre-existing ones. It differentiates between different types of biochemical activities for de novo emerging sequences, and, rather than theorizing upon the legitimacy of what function should mean, it decomposes this complex concept into a hierarchical organization of properties of the object in time and space. It can be seen as a conceptual tool well adapted to describe molecular objects (DNA loci, RNA transcripts and proteins) in a manner that roughly maps to subfields of research and associated measurement techniques.”

- Explaining how Evolutionary Implications relate to Selected Effect but can describe many types of selection and other evolutionary processes:

“The meanings of Physiological Implications and Evolutionary Implications are deliberately broad to allow for detailed descriptions beyond the somewhat restrictive notion of selected effect. For instance, Evolutionary Implications can refer to selection upon a trait driven by the object in the past, as in selected effect (Millikan, 1989), but it can equally describe novel adaptive effects, as one may expect in gene birth (Carvunis et al., 2012), or other types of influence on population dynamics such as runaway selection.”

- Defining the meaning of the arrows

In the legend of Table 1:

“Table 1: The Pittsburgh model of function – definitions and relationships. Vertical arrows: necessary but not sufficient. For example, an object with Physiological Implications may not have Evolutionary Implications. The hierarchy of the arrows is an interpretation that did not directly derive from our textual analysis”

In the text where we describe the model:

“Conveniently, these five meanings map to a general interpretation of the logical flow of genetic information over time and space. Starting from an object’s presence (Expression), we consider its physical properties (Capacities), binding partners within a system (Interactions), phenotypic impact (Physiological Implications) and influence on population dynamics (Evolutionary Implications). We propose to relate these five meanings of function in a logical hierarchy whereby existence of a given meaning of function is necessary, but not sufficient, to infer existence of a higher meaning. For instance, as has been discussed in the context of the ENCODE controversy, Expression is necessary but not sufficient to cause Evolutionary Implications (Doolittle, 2018). In other words, this model acknowledges the spatio-temporal relationships between these concepts while enabling researchers to study them independently of each other.”

And after the experimental example:

“Evidently, neither technique gives direct insights into Evolutionary Implications since Expression and Interactions are necessary but not sufficient to cause Physiological Implications, let alone Evolutionary Implications.”

- Explicitly stating why Figure 1 shows that the literature is confusing:

“In other words, the model gave readers a key to decipher more specifically what authors meant by the word function, but the same instance of the word in the same sentence was most of the times (37/42 = 88%) interpreted differently by our team members.”

3) Cancer example and usage of function:

Reviewers noted issues with the way we presented a theoretical example where the use of function is problematic, and pointed out that, in this example and in general, one should pay attention to the grammatical context of function. We are grateful for these criticisms, which have led to novel analyses and more subtle re-writing in the revised manuscript. The specific reviewer comments are listed below and our response follows.

Reviewer 1:Authors do problematize something like this, though having a de novo gene that causes cancer seems to be over-problematizing. I think many who would say "Gene X functions in the development of cancer [or even some disease that does not itself evolve]" would balk at saying that "The function of gene X is to cause cancer".Reviewer 2:12) Subsection “A model of function for de novo gene birth research”. The example seems contrived. Why conflate the concept of de novo gene birth with the fact that many disease relevant mutations are not in selected elements. These should be two separate examples. Ex1: mutations that lead to disease but lie in non-conserved regions. Ex2: gene birth example.Reviewer 3:5) In their rhetorical analysis, the authors have considered that the meaning of "function" is similar when it is used as a name ("assessment of function"), a verb (MDF1 functions in two important pathways")), and adjective ("functional")). However, some philosophers have pointed out that "a function", "to function", and "functional" have different meanings, expressed by different words in other languages than English. For example, in the cited though experiment, scientist who may be reluctant to say that "the function of the pro-oncogenic gene is to generate cancer", but agree to say that "the function of the anti-oncogenic gene is to prevent cancer", would agree to say that both genes are "functional". The authors should provide the numbers of instances of the different words used and justify that there is no different meanings statistically associated to the name, the verb and the adjective.

We are grateful for the reviewers’ suggestions to consider the grammatical nuances of function usage in our database of abstracts and as they relate to our thought experiments. Our revised manuscript includes a detailed breakdown of the number of times we saw function used as a noun, verb, adjective and adverb:

In the main text:

“Each member of our team independently assigned one or more meaning to each instance of the word function found in our database of abstracts (4 members; 20 abstracts; 42 instances including 25 nouns, 12 adjectives, 3 verbs and 2 adverbs; see Approach).”

And in the approach:

“Forty-two usages (25 nouns, 12 adjectives, 3 verbs and 2 adverbs) were analyzed across 20 abstracts.”

The details of which instance was used in which grammatical context and assigned which meaning(s) is now included in Figure 1—source data 1, along with a complete description of all our results.

We did not detect notable usage differences between grammatical contexts, with the exception of “vague”, which when in a consensus assignment was almost always a noun. We added this interesting result in the revised manuscript:

“We did not observe strong differences in how function was interpreted when it was used as a noun or an adjective, but such differences may become detectable in a larger text sample. Interestingly though, all but one instances with a vague consensus assignment were nouns.”

We agree with the criticisms raised by all three reviewers about how we phrased our thought experiment. In the revised manuscript:

- We simplified it using “deadly disease” instead of cancer, which over-problematized things as noted by reviewer 1.

- We stratified it in a manner inspired by reviewer 2 that separates the de novo question (is it a gene?) and the disease question (does it matter?).

- We clarified that, independently of whether and how it impacts physiology, a de novo sequence cannot have a selected effect function as suggested by reviewer 3.

- We use the example to illustrate the richness of grammatical nuances suggested by reviewers 1 and 3.

- We only mention it once in the revised manuscript (instead of twice in the submitted manuscript).

“The meanings of function are at the heart of what constitutes a de novo gene birth event. For a genomic sequence to be labelled as a gene, it must by definition have a function. If such a gene has evolved de novo, the locus it came from by definition was not a gene, thus did not have a function, or at least not a function of the same nature as the one the new gene has. The molecular objects of study are thus transitioning between a state without a function and a state with a function. They cannot have a function by the selected effect definition, since their existence cannot be explained by a past selection. The transformative nature of the de novo emergence process thus renders the debates about when a locus actually becomes functional highly contentious.

Let us consider a hypothetical example to illustrate the difficulty of thinking about the function of recently emerged coding elements. Imagine a locus that has become transcribed and translated for the first time in somebody’s gamete, generating a novel protein whose expression propagates to future generations. The corresponding protein has no activity whatsoever. Does this locus correspond to a de novo functional gene? Not according to the selected effect definition, since there is no past selection to explain why the locus is here. Not according to the causal role definition either, since it does not do anything. What if the protein is adaptive? Still, the locus would not have a function according to the selected effect definition, although it might be in the process of acquiring one. What if the protein causes a deadly disease? In that case many may find it acceptable to use derivatives of function and write “the locus functions in the development of disease”, or “the locus is functional”, but not “the function of the locus is to cause a deadly disease” (Doolittle et al., 2014). One might cautiously call this locus a proto-gene (Carvunis et al., 2012), Zombie or Lazarus DNA (Graur et al., 2015), rather than a functional gene. But, at any rate, this locus “matters” (Ardern, 2018) and there is a pressing need to find the right words to express what it does, why and how.”

Reviewer #1:

We thank the reviewer for these positive comments, keen suggestions and questions, which have helped us greatly improve our manuscript. Part of them are addressed in the grouped section of our letter, and the rest is addressed below.

Introduction. The first sentence of this manuscript makes a very strong claim – not just that scientists inevitably express results in words, but that they do so "in terms intended to persuade or impress". At least that's how the OED in my Mac defines "rhetorical" and it's what scientists try NOT to do, when describing observations at least. Perhaps we never succeed in being "objective", and of course we do want to present theories in the best light, but the goal of science is NOT to "intend to persuade" in presenting data. Do authors mean to claim we never succeed in getting rid of such an intent? The second paragraph reminds us, usefully, that words mean different things to different people, which I grant – just not that any use of words is necessarily "rhetorical".

The reviewer is of course right that *any* use of words is not necessarily meant to persuade or impress. Yet the first sentence of our manuscript asserts that the scientific enterprise as a whole is rhetorical. Here the adjective “rhetorical” means “of, relating to, or concerned with rhetoric” (Merriam-Webster), referring to the academic discipline of rhetoric as illustrated by the three references at the end of the sentence. Rhetoric is the art of writing (or speaking) effectively. It includes persuasion but more broadly treats with argumentation and composition. Even though scientists generally strive for objectivity, every manuscript is a composition where data and theories are presented within a carefully crafted context chosen by the authors. As a result, even if an individual scientist’s experiments and models follows the ideal of pure scientific objectivity, science as a global enterprise is a conversation where facts and theories contribute to evolving argumentations constructed by different people. It follows that science is inevitably influenced by socio-cultural parameters. We also often rely on figures of speech and metaphors (“gene birth”) that shape how we think about problems. This is an important difference from the definition of “rhetorical” given by this OED, which rhetoric scholars have criticized as “merely rhetorical” because it understates the larger rhetorical processes at play.

Introduction. I'm a little unsure why references to me (Doolittle) include my initials or middle name (Ford) while others just have the last name only. Ford is not part of my surname.

Thank you very much for catching this glitch in our reference manager, we have corrected it so that only your last name appears in the citations.

Introduction. I think it is (or should be) more correct to say that ENCODE critics like me don't so much insist that everyone use the SE definition of function, as that they be more precise about the definition they ARE using. Many genomicists and molecular biologist seem to think the "function" is unproblematic. This might be so if all activities arise by selection, and I think that most genomicists and molecular biologists are indeed pan-adaptationists, although not realizing it! Authors do say that something like this in the second paragraph on this page.

We fully agree and have edited our revised manuscript accordingly:

“Yet some argue that the evolutionary meaning of function should be the only relevant one for DNA sequences” (Graur et al., 2013), and that when referring to other meanings scientists should use other words such as “effect” or “activity” (Doolittle et al., 2014)”

We also agree with the notion that many genomicists are unknowingly pan-adaptationists and address this important point further in our response to reviewer 3’s questions about confusion (points 4, 6 and 7).

Introduction. I like the idea of thinking about the "function" of newly arising genes. I think I agree with them that the idea of a gene implies a function, but wonder whether one would say, of a newly arisen stretch of DNA with a promoter, open reading frame producing a substantially long protein with no activity whatever, and terminator, that it was not a gene? One could easily make such a thing in the lab. Would we not call it a "gene", or would we also require that it has a biological function?

Artificial coding sequences have been made in the lab, but they are usually not called genes, even if they can have drastic effects on phenotypes and fitness which, if they were observed of natural sequences, would probably be deemed “functional” (for example Knopp et al., 2019; Neme at al., 2017). Hence, while the concept of gene implies indeed a function, a function that is measured in the lab for an artificial sequence does not necessarily imply that sequence deserves the status of gene, at least for some authors. The relationship between gene and function thus does not appear to be a strict equivalency. This raises very interesting follow up questions and could be the focus of another rhetorical study in the future. In our revised manuscript, we now discuss the case of a natural sequence has no activity (see grouped answers, #3: Cancer example and usage of function). An interesting aspect of our model is that a natural or artificial sequence that does not do anything can still be described in terms of its Expression and Capacities, although we do not discuss it explicitly.

Not altogether sure what they mean "hinder scientific development". They say (and I agree) that "function" and "matter" (as in "Does it matter?") are often conflated, but possibly any further definition of function will do.

We meant hinder scientific communication, which in turn slows down the pace of scientific discovery because results are not interpreted as authors intend them to be. In this manuscript we show that communication is indeed affected by the use of function in the field of gene birth. Indeed, any further definition would do, we agree – and the one we proposed in our model may possibly remedy the issue. We have replaced “development” by “communication” throughout our revised manuscript to increase clarity.

Subsection “A model of function for de novo gene birth research”. Pittsburgh model seems very useful and appropriate. Was it just invented for this paper, or does it have a provenance? I think the former, but this should be made clear.

We thank the reviewer for these compliments about our model. We did build it in this paper and have clarified this in the revised manuscript by modifying the subtitle of this section of the paper from “A model of function for de novo gene birth research” to “Constructing a model of function for de novo gene birth research.”

Subsection “The multiple meanings of function hinder scientific development in the field of de novo gene birth. Not sure what "consensus" means. Did all authors get together and try to agree (which is what I think the term usually means) or did they just take the majority of independent opinions.

We discussed and tried to agree on meanings. This is indicated in the sentence:

“we collectively reviewed and discussed all independent assignments for each instance of the word and built a consensus set of assignments”

And described in the Approach section. To further increase clarity, we have indicated this also in the figure legend of Figure 1:

“B – Distribution of the number of meanings per instance after consensus agreement by the readers. Consensus was built through conversations and agreement between the authors, rather than majority opinion. In one case, consensus could not be reached and three meanings were assigned to reflect all the differing interpretations of our team members.”

Subsection “Interpretation and recommendations”. Totally agree that when the word "function" is used, it should be qualified, or else it should not be used.

Indeed, and in the revised manuscript we now cite one of your pieces on this exact subject at the relevant sentence of this page:

“Rather than privileging one meaning of function over another, we endorse qualifying the use of function or avoiding the word altogether (Doolittle, 2018).”

Reviewer #2:

We are grateful for this reviewer’s comments which we have addressed below and in the grouped response section of our letter.

Since several of the points raised relate to tone, which the reviewer felt was less than scholarly, we also want to clarify that this manuscript was written this way on purpose to adhere to the “active/engaging” style guidelines of *eLife* feature articles, and to be as accessible as possible to an interdisciplinary audience. These efforts are acknowledged by reviewer 4 (“*It is particularly notable how accessibly the manuscript is written for a multi-disciplinary audience***”**). We regret that this reviewer did not appreciate this and have addressed each of their specific criticisms in our revised manuscript.

3) Introduction: The paper frames itself in the context of ENCODE, and then shifts gears and focuses on the field of gene birth. The authors should take a broader perspective of the utilization of the word function in their Introduction, or take a much narrower perspective focusing on gene birth. The current Introduction distracts instead, and feels more like a bait-and-switch. If the authors choose to start with ENCODE in the Introduction, they should instead take on the task of classifying the 300-some papers by the ENCODE consortium instead.

We understand the reviewer’s concern, which we believe arose from the fact that our submitted manuscript did not make it abundantly clear that the study of gene birth can be considered as a subset of genomics (although it is of course interdisciplinary). The encode controversy serves as an effective illustration of the general problem of function in genomics, before diving into the more specific edge-case of de novo gene birth. We do not see this structure as a shift per se, but rather an increased focus on a specific sub-question. We have revised the Introduction of our manuscript to clarify that we are concerned about function as it relates to genomic objects:

“In brief, the conversation divides those who argue function should mean why an entity does what it does (the selected effect definition of function) and those who argue it can also mean what an entity does (the causal role definition of function) (Laubichler et al., 2015). Other theories of biological function have been proposed, including an organizational account of function (Mossio et al., 2009; Roux, 2014), but the selected effect / causal role perspectives have dominated the conversation in the context of genomics.

Far from a fruitless dispute over semantics, the rhetorico-scientific debate has sparked a number of thoughtful studies that have advanced thinking at the interface of evolutionary biology and genomics”.

And:

“We focused our attention on a relatively recent subfield of evolutionary genomics studying the specific case of de novo gene birth.”

We are reluctant to include more introductory examples of function beyond genomics, such as describing the function of traits, or the function of organs, as we believe that our approach and reasoning really might be relevant only for the specific type of biological objects studied in genomics (DNA, RNA, Proteins).

7) Introduction: The first sentence of the third paragraph suggests a very narrow view of the debate surrounding the ENCODE project and the definition of function therein. This is probably not an area that the authors want to get into. In particular, searching for "ENCODE debacle" in Google Scholar returns only one paper, which has never been cited. Searching in Google returns only blog posts by Dan Graur and other angry bloggers, which is probably not the view that the authors want to align themselves with. A closer reading of the original ENCODE 2012 paper provides upfront a definition of "biochemical function" (which perhaps should be referred to as "biochemical activity", despite again the unfortunate naming of "functional assay" and "functional genomics"). It then uses that definition, and very clearly indicates that only a small fraction of the genome is under evolutionary selection. Even the criticisms of ENCODE have primarily cited the press articles written by news authors, not scientists, that claim that 80% of the genome is functional. Anyways, I would skip that whole section if the authors don't want to re-open a very large can worms.

We thank the reviewer for pointing out the problematic use of “debacle”, and that our summary of the encode problem did not reflect what the encode scientists wrote specifically enough. We have rephrased the corresponding paragraph in the revised manuscript, so that we use the term “controversy” rather than debacle to avoid “angry blogger” connotations and we represent the encode announcement more specifically:

“A spectacular example of how conflict arises from differing interpretations has become known as the “ENCODE controversy” – a heated scientific debate over what fraction of the human genome is “functional”. Is it approximately 80%, as suggested by biochemical evidence, or is it closer to 10%, as it appears from evolutionary evidence (ENCODE Project Consortium, 2012; Doolittle, 2013; Graur et al., 2013; Kellis et al., 2014)?”

This characterization of the 2012 ENCODE announcement closely matches the language used in their abstract:

“The human genome encodes the blueprint of life, but the function of the vast majority of its nearly three billion bases is unknown. The Encyclopedia of DNA Elements (ENCODE) project has systematically mapped regions of transcription, transcription factor association, chromatin structure and histone modification. **These data enabled us to assign biochemical functions for 80% of the genome**, in particular outside of the well-studied protein-coding regions. Many discovered candidate regulatory elements are physically associated with one another and with expressed genes, providing new insights into the mechanisms of gene regulation. The newly identified elements also show a statistical correspondence to sequence variants linked to human disease, and can thereby guide interpretation of this variation. Overall, the project provides new insights into the organization and regulation of our genes and genome, and is an expansive resource of functional annotations for biomedical research.” (emphasis ours)

8) Introduction: The selected effect that gave rise to a trait or a genomic region may be quite different from the current functional roles of that trait or genomic region. Thus, equating function to "selected effect" may be inaccurate as well.

We agree with the reviewer, the Introduction summarizes the literature and does not reflect our view.

9) Introduction: "scientists cannot agree on the number of functional genes in the human genome" is a peculiar statement to accompany the Pertea and Jungreis papers. Briefly, Pertea claims to discover thousands of new genes, Jungreis claims that Pertea made specific mistakes resulting in exclusively false positives. Yes, there is debate, but the Pertea paper is not a reference for this statement. This sentence alone indicates a lot of nonchalance on the part of the authors about dismissing the state of broad fields that they should be much more cautious about, especially in a paper that seeks to bring rigor to the field.

We agree with the reviewer that “Jungreis claims that Pertea made specific mistakes resulting in exclusively false positives”. We chose this example to illustrate the debate, because some of what Jungreis identifies as mistakes made by Pertea relate directly to the question of function. Indeed, Pertea’s claims were made based on expression, and Jungreis’ claims were made based on selected effect. This can be seen by contrasting both abstracts:

“We assembled the sequences from **deep RNA sequencing experiments** by the Genotype-Tissue Expression (GTEx) project, to create a new catalog of human genes and transcripts, called CHESS. The new database contains 42,611 genes, of which 20,352 are potentially protein-coding and 22,259 are noncoding, and a total of 323,258 transcripts. These include 224 novel protein-coding genes and 116,156 novel transcripts. We detected over 30 million additional transcripts at more than 650,000 genomic loci, nearly all of which are likely nonfunctional, revealing a heretofore unappreciated amount of transcriptional noise in human cells. The CHESS database is available at http://ccb.jhu.edu/chess.” (Pertea, emphasis ours).

“In a 2018 paper posted to bioRxiv, Pertea et al., presented the CHESS database, a new catalog of human gene annotations that includes 1,178 new protein-coding predictions. These are based on evidence of transcription in human tissues and homology to earlier annotations in human and other mammals. Here, we reanalyze the evidence used by CHESS, and find that nearly all protein-coding predictions are false positives. We find that 86% overlap transposons marked by RepeatMasker that are known to frequently result in false positive protein-coding predictions. More than half are homologous to only nine Alu-derived primate sequences corresponding to an erroneous and previously withdrawn Pfam protein domain. **The entire set shows poor evolutionary conservation and PhyloCSF protein-coding evolutionary signatures indistinguishable from noncoding RNAs, indicating lack of protein-coding constraint.** Only four predictions are supported by mass spectrometry evidence, and even those matches are inconclusive. Overall, the new protein-coding predictions are unsupported by any credible experimental or evolutionary evidence of function, result primarily from homology to genes incorrectly classified as protein-coding, and are unlikely to encode functional proteins.” (Jungreis, emphasis ours).”

Please note that we do not intend to take sides, simply to highlight that defining what a “gene” is depends on what a “function” is.

10) "Introduction, "practically zero". This is again a great oversimplification of a rich field of gene birth, and does not reflect well on how scholarly the authors should be.

This 1977 quote, and the sentiment it expresses, are often cited as an explanation regarding why the existence of de novo genes was traditionally considered implausible and at best anecdotic. As we explain in the next sentence, the field had to wait until the explosion of genomics to come to terms with the idea that this process may in fact be a major contributor to evolutionary innovation. The mechanisms of de novo gene birth are not yet understood. We feel our short summary of the field is appropriate for the scope of this manuscript. However, in the revised version of our manuscript, we include the most recent and most complete review of the field so that readers can learn more if they are interested, and also so that the expertise of Carvunis (last author of the review and this manuscript) is more evident:

“Studies of a growing number of individual gene candidates have confirmed their de novo emergence, fueling many genomic and evolutionary studies to evaluate the scale and mechanisms of the de novo gene birth phenomenon (Tautz and Domazet-Lošo, 2011; McLysaght and Hurst, 2016; Van Oss and Carvunis, 2019).”

11) Introduction: "gene". Why choose this word, rather than "functional". The word "gene" itself has a long history of differing definitions and great debate. Once more, throwing this word around without much thought seems out of place for a paper that seeks to be scholarly.

We have incorporated the reviewer’s suggestion in the revised manuscript:

“The transformative nature of the de novo emergence process thus renders the debates about when a locus actually becomes functional highly contentious.”

13) Subsection “A model of function for de novo gene birth research”: Please reword to avoid the word "fantastic".

We have incorporated the reviewer’s suggestion in the revised manuscript:

**“**All these challenges make the field of de novogene evolution a well-suited test bed […]”

15) Subsection “A model of function for de novo gene birth research”: Naming this Pittsburgh after the affiliation of the last author seems inappropriate. Does every scientist in Pittsburgh agree? Why not "Carvunis", since she's the only author from Pittsburgh? Why not "our" model, and let others name it "the Carvunis model".

We chose to name the model after Pittsburgh because the model crystallized when we all were together in Pittsburgh for a collaborative retreat. We have added more information about why we chose this name in the revised manuscript:

**“**This work resulted in a structured classification of the meanings of function specifically adapted to the de novogene birth literature, which we named the Pittsburgh model of function after the geographical location where the model crystallized at the occasion of a collaborative retreat between our team members (Pittsburgh, PA, USA).”

18) Table 1: E: does mere presence of DNA make every DNA segment functional?

For a DNA object, expression refers to the presence or amount of its transcription or translation product, as per the definition written in Table 1:

“The presence or amount of the object under investigation (RNA or protein object), or the presence or amount of its transcription or translation products (DNA object)”.

20). Roux, 2014 also brings up many additional important points worth discussing at greater length.

We wholeheartedly agree that Roux, 2014 makes a lot of important points. We are delighted that Dr Roux, the author of Roux, 2014, also reviewed our submitted manuscript (reviewer 3). We trust that having addressed all of Roux’s specific comments simultaneously addresses this suggestion to discuss Roux, 2014 even more.

21) Subsection “The multiple meanings of function hinder scientific development in the field of de novo gene birth”, "only 12% were unanimous":- which ones were most confused and why- this result perhaps indicates that nomenclature would not have helped, may be either too ambiguous or overspecified.

We thank the reviewer for asking which of the non-unanimous assignments were the most confused. We inspected the top 3 most confused instances, with 4 or 6 different assignments before agreement, and found that they all included the “vague” category, and vague was ultimately assigned in the consensus. A speculative interpretation of why the most confused instances were vague may be that the coders attempted their best to interpret a term that was indeed used vaguely, resulting in multiple possibly random assignments. We have revised our manuscript to include this interesting observation:

“In other words, the model gave readers a key to decipher more specifically what authors meant by the word function, but the same instance of the word in the same sentence was most of the times (37/42 = 88%) interpreted with more than one meaning by our team members. The most confusing instances, with 4 or 6 distinct assignments, were all assigned”vague” by at least one reader.”

Our result that only 12% of initial assignments were unanimous cannot, on its own, indicate whether our nomenclature would have helped, since it was not used in the texts. It is absolutely certain that, had the authors written their abstracts according to our nomenclature, there would have been no ambiguity since the term function would not have been used or it would have been contextualized. We hope the future will tell if indeed our nomenclature will help de novo gene birth scientists communicate more effectively.

22) Subsection “The multiple meanings of function hinder scientific development in the field of de novo gene birth”. "again supporting our hypothesis…". Perhaps also suggesting that the proposed classification scheme does not work?

To address this comment, we have removed the clause in question in the revised manuscript:

“This shows that, in our analyses, 21/42 (50%) instances were interpreted differently by at least one of four readers upon independent reading of the texts”

Reviewer #3:The authors address an interesting question regarding the philosophical issue related to the meaning(s) the word "function", and in particular the question whether a selective process is required to legitimate the usage of the word "function". Though extensive philosophical debate has occurred within the last decades, little attention has been paid to the usage of the word by the biologist themselves. The birth event of de novo gene, understood as the event that occurs prior to any selective process, is an interesting limit case to see whether biologists use or not the word "function" to characterize some particular properties of these genes. However, as it is presented in the manuscript, the work exhibits several conceptual and methodological weaknesses.

We thank the reviewer for recognizing the novelty of our study of how biologists use the word function in the context of gene birth, and for his constructive comments. We have addressed them all and feel our revised manuscript is much improved thanks to this input.

A) The philosophical issue of de F-word:2) Regarding the teleological-etiological theories, in parallel with the selected-effect theories of function, a new approach has emerged in the last decade, the so-called organizational approach of function (OAP). For these theories, what a function means is, as for the selected effect theories, "Why is it here?", but the legitimation is not grounded on a selective process but on organizational properties, basically the existence of a causal loop between the causal role of a trait in the self-maintenance of the system, and the production and maintenance of a trait by the system. The application of this theory to the case of de novo gene birth is interesting, but there is no reference to this theory of function in the manuscript.

We wholeheartedly agree with the reviewer that applications of the OAP theory to gene birth would be very interesting. This theory was not directly mentioned in our submitted manuscript because it has not been explicitly applied to genomics. It would indeed be very interesting to do so, and to focus on de novo genes in this context, but we feel such philosophical innovation would be beyond the scope of our study. That said, we have revised our manuscript to include mention of the OAP theory in hopes it will inspire philosophically inclined readers to think about it:

In the Introduction we added a sentence:

“In brief, the conversation divides those who argue function should mean *why an entity does what it does* (the selected effect definition of function) and those who argue it can also mean *what an entity does* (the causal role definition of function) (Laubichler et al., 2015). Other theories of biological function have been proposed, including an organizational account of function (Mossio et al., 2009; Roux, 2014), but the selected effect / causal role perspectives have dominated the conversation in the context of genomics.”

In the approach we also specify that the SE/CR question has been addressed previously in genomics:

“The need for an improved model of function

We began the qualitative analysis by establishing the need for refining the selected effect / causal role binary model discussed in the philosophical literature of genomic function.”

3) Taking into account the philosophical issue of the F-word, the emergence of de novo genes is problematic only for the supporters of the selected-effect theories; it is not for the advocates of the causal role theory and the OAP. Indeed, the main critic of the OAP against the selected effect theory is precisely that this theory cannot account for the emergence of function. The title is hence rather misleading, since it refers to a privileged theory of function. The word "evolution" is, in the same idea, also confusing. Why use it rather than emergence? Indeed, if the de novo emergence of gene "function" has anything specific regarding the F-word philosophical issue, it is that it is de novo, i.e., it has not (or not yet) been submitted to an past selective pressure. If the existence of de novo genes and their effects can be interpreted as a consequence of a selective pressure, they are no longer "de novo". The meaning of "evolution", used for de novo gene, should be precised. If it precisely means the birth event of a new gene, this event occurs prior to any selection, because selection can act only by differential screening between already existing traits. In this meaning, "emergence" seems more adequate.

We thank the reviewer for pointing out that evolution is not the most appropriate term here. We agree and have revised our title accordingly: “The meanings of “function” in biology and the problematic case of de novo gene emergence”. Throughout the manuscript, we use birth and emergence interchangeably.

We agree with the reviewer that the selected effect definition of function is the most problematic in the context of gene birth. This is clarified in our revised manuscript:

“The meanings of function are at the heart of what constitutes a de novo gene birth event. For a genomic sequence to be labelled as a gene, it must by definition have a function. If such a gene has evolved de novo, the locus it came from by definition was not a gene, thus did not have a function, or at least not a function of the same nature as the one the new gene has. The molecular objects of study are thus transitioning between a state without a function and a state with a function. They cannot have a function by the selected effect definition, since their existence cannot be explained by a past selection.”

However, the causal role definition is problematic too. What evidence would one require to declare that a locus has, or has not, a “causal role” function, if that were how one would declare a gene to be born? As pointed out for example by Doolittle, 2014 and in our Approach section, the philosophical concept of causal role has not become widely adopted in the scientific practice because it is difficult to define what would be a relevant/meaningful causal role. Is expression enough? Interactions? Do we need to see a deletion phenotype at the organismal level? An interesting aspect of our model is that a locus can be described in terms of expression, capacities and interactions, even if there is no impact at the whole organism level. In the revised manuscript, this is now further clarified in the thought experiment we use to illustrate the difficulty of thinking about function in the context of gene birth (please see #3: Cancer example and usage of function).

Altogether the philosophical issues compound into practical problems of language. This is the main point of our manuscript, which we believe justifies the adjective “problematic” in our title.

B) semantic plurality of the F-word and conceptual confusion.4) The authors claim that the existence of several meanings of "function" is a source of confusion between biologists, but there is no reported evidence of such a confusion. Actually, the word "function" is used in biological sciences at least from the sixteenth century, without particular debates within the biologist about what this word means. The fact that there is a philosophical debate about the legitimacy of the meaning of a notion like "function", and the fact that this notion is a useful or a confusing conceptual tool is 2 different questions. The authors should precise if the object of their work is to contribute to the philosophical “debate by clarifying the meaning of the concept by the biologists, or to highlight to existence of real confusion or discrepancies in the usage of the word by the biologists, specifically in the field of de novo gene research.

Thank you for this excellent question. Our work both highlights the existence of real confusion and contributes to the philosophical debate.

Real confusion in interpretation of the word is shown in the poignant disagreements between independent readers for the same usages (Figure 2A), the multiple meanings found in our small text sample (Figure 2C), and in the fact that even after discussions several meanings are still assigned for many usages (Figure 2B). Our work thus exhibits the ambiguity of the use of the concept of function as it relates to the field of gene birth. Our result is therefore very novel since, as the reviewer points out, there has not been much debate among biologists about what function means previously. There are notable exceptions, including by reviewer 1 whom we cite.

We believe reviewer 1 explains very well in one of his comments why there has not been much debate although the concept really is confusing:

“Many genomicists and molecular biologist seem to think the "function" is unproblematic. This might be so if all activities arise by selection, and I think that most genomicists and molecular biologists are indeed pan-adaptationists, although not realizing it!”

In other words, function is not perceived as confusing because many members of this field consider, to put it bluntly, that if a locus is expressed then it does something and that thing matters. This is of course not true given a modern understanding of evolutionary theory, nor given the discoveries regarding pervasive expression and promiscuous binding that have been made in the past couple of decades. But the implications of these discoveries for the concept of function have not yet fully penetrated the field. We hope our manuscript will help in this regard.

We have revised our manuscript to highlight these two aspects of our work:

“In summary, our results provide quantitative support to previous assertions (Doolittle, 2018; Laubichler et al., 2015) that the word function carries convoluted meanings that complicate the scientific conversation. Our rhetorical analysis shows that function becomes an ambiguous concept when applied to the edge case of de novo gene birth, which leads to confusion in interpretation of the literature”.

The philosophical contribution of our manuscript is not to clarify “the” meaning of the concept by biologists. Indeed, we instead show that the concept has multiple meanings in its current usage of this field. This contributes to the philosophical debate by showing a discrepancy between prevailing theories of genomic function and the way the word is circulated in practice. Our work also contributes to the philosophical debate by suggesting that there may not be a single legitimate meaning of function – many might exist, and they might be field-specific. We have clarified this in our revised Discussion section:

“Function is a concept that depends on the methodological practices, measurement procedures and habits of scientists. As these change, so too will the concept of function change and adapt to specific subfields of research, because it is contingent and always in a state of flux.”

Our data could be used by a philosopher of science as one starting point to tackle the questions of legitimacy, since these questions have to deal with questions of use.

6) The existence of different meanings of a notion does not necessarily imply that the notion is conceptually confusing, if the different meanings are complementary and not contradictory. Actually, the fact that the authors have been able to attribute, according to the "Pittsburg model of function", identified meanings to the usage of the term "function" can be interpreted as a proof that the word is not so confusing.

We would like to clarify the difference between whether the concept is confusing, and whether interpretation of the literature is confusing. We have addressed the former in answer to this reviewer’s point 4. But when in our manuscript we used the word “confusion”, we were referring to the latter meaning. For example, reviewer 2 asked us to list the “most confused” examples (we found that those instances assigned the most different meanings by independent readers were all found “vague” ultimately). Reviewer 1 used also the term confusion with the same meaning in Doolittle et al., 2014. In any case, to prevent such a possible misunderstanding, we have clarified what we meant in our revised manuscript:

“The general confusion about what exactly is meant by “function” across the literature is such that some have pleaded for the community to deal with the “F-word” urgently (Doolittle et al., 2014; Doolittle, 2018).”

With this understanding of interpretative confusion, we hope that it is clear from Figure 2A that function is confusing since each independent reader tends to interpret it differently within the same sentence. We have clarified this in our revised manuscript:

“In other words, the model gave readers a key to decipher more specifically what authors meant by the word function, but the same instance of the word in the same sentence was most of the times (37/42 = 88%) interpreted differently by our team members.”

Including mention of a case where consensus could not be reached:

“In one case, consensus could not be reached and three meanings were assigned to reflect all the differing interpretations of our team members.”

8. Regarding the usage of the word "function", the main conclusion of the rhetorical analysis is that the large majority of the meanings are physiological implications and evolutionary implications, which represent almost all of the instances (40 on 42). The authors should discuss the implications of such a rather "integrative" or "high level" meaning of function on the philosophical debate about what a function mean. The fact that, as stated by the authors, "evidently, neither technique gives direct insights into evolutionary implications" should also be discussed. Do the authors consider that this methodological difference impacts the legitimation of the meaning of function?

We thank the reviewer for these observations and questions.

The main conclusion of our quantitative analysis is rhetorical rather than philosophical: the word function can mean different things, sometimes different things at once, and this makes it difficult to understand the literature. We have clarified this in our revised manuscript:

“Yet our rhetorical approach leads us to conclude that the literature in this field is hard to interpret, and that further nuance in the use of function would assist the reader in understanding how the authors intend their results to be interpreted.”

Our manuscript is aimed at illustrating the plurality of meanings in the usage of the word function in the current scientific practice rather than legitimation of such meanings. Studying whether and how methodology impacts legitimation seems beyond the scope of our present study but represents a very interesting question for future direction. We do not hold an opinion on this question at this point. In the revised manuscript, we further discuss how methodology relates to meaning in our model:

“Evidently, neither technique gives direct insights into Evolutionary Implications since Expression and Interactions are necessary but not sufficient to cause Physiological Implications, let alone Evolutionary Implications.”

The reviewer is correct that the large majority of instances are EI or PI, but they represent 33/42 rather than 40/42 instances. This is because EI and π can be found together assigned to a single instance (6 times), and also because we had made a typo in the submitted manuscript. These numbers are now clearly stated in the revised manuscript:

“The meanings most frequently found in our consensus assignments were Physiological Implications (21 instances) and Evolutionary Implications (18 instances) (Figure 1C). These two were also most often found assigned together in cases where a single instance was assigned two or three meanings (6 instances).”

We do not feel confident making philosophical claims based on these numbers alone, since it could simply reflect that these terms are too ambiguous, as suggested by Reviewer 2, or that biophysicists are only beginning to study gene birth.

C) Methods10) Another problem is how the abstracts have been selected. A sample can be considered as representative only if it have been randomly picked up, which is not the case in the study. At the best, it can be considered as illustrative. Is the fact that the person who selected the abstracts is an "expert in the field" supposed to be, by itself, a guarantee of objectivity?

We thank the reviewer for pointing this out. Our set of abstracts is diverse but it is not, in fact, “representative”. We have removed this sentence from the text, leaving the clear explanation of how abstracts were selected:

“A library of 20 published papers that included the term function or its derivatives (functional, functioning) in the abstract was assembled by a team member who is also a published expert in the field of de novo gene birth (Table 2). Publication dates span from 1992 to 2017, with most dated after 2012 because this is a recently expanding field. The papers were chosen to span a variety of journals, countries, citation counts, model organisms, methodologies and scope, in order to derive a context specific rhetorical argument (McGee, 1990). Our library is estimated to represent ~2% of the literature published on the topic of de novo gene emergence, as a Google Scholar search returns 972 results in December 2018 (‘‘"de novo gene birth" OR "de novo gene evolution" OR "de novo gene emergence" OR "de novo genes"‘).”

11. Another important methodological limitation is the fact that the sample of abstracts has been used to build the model, to validate it, and to use it is basically the same; indeed, 17 of the 20 abstracts where used for model refinement, and the remaining 3 for its validation. After that, the same 20 abstracts where used for quantitative analysis and "hypothesis testing". I do not clearly see what are the tested hypotheses, but the fact that the same set of material and the same set of coders where used all along the process it for me a serious methodological limitation.

We agree with this criticism and, following editorial suggestion, have rephrased our revised manuscript to remove mention of hypothesis testing. The specific edits made to address this are listed below.

“With this model in hand, we analyzed whether the multiple meanings of function impact understanding of the literature in the field of de novo gene birth.”

“This quantitative analysis shows that the unspecified multiple meanings of function confuse the literature”

“This shows that, in our analyses, 21/42 (50%) instances were interpreted differently by at least one of four readers upon independent reading of the texts.”

“Finally, we performed a quantitative content analysis to analyze how the multiple meanings of function affect understanding of the literature in the field (Neuendorf, 2016).”

“Quantitative analysis”

**“**We used the Pittsburgh model of function to analyze whether the unspecified multiple meanings of function hinder understanding of the literature in the field of de novo gene birth.”

We hope that our work, which represents the first quantitative analysis of how function is used in scientific abstracts, can be used as a stepping stone upon which hypotheses can be constructed and rigorously tested in the future.

Reviewer #4:General assessment: this manuscript is an important and thoroughly argued contribution to eLife. It is particularly notable how accessibly the manuscript is written for a multi-disciplinary audience. The arguments are timely, and the need for this work is justified in the opening section. The description of the findings and later description of the approach are detailed and explanatory, with interpretation provided as relevant. I highly recommend this piece for publication.

We are grateful to the reviewer for these positive comments and recommendation.

[Editors' note: further revisions were requested prior to acceptance, as described below.]

Reviewer #1:I still wish that the authors had looked at more than 20 abstracts, but the writing is very clear and the intellectual clarification this promises is enough, in my view, to justify publication. And I'm sure that many in future will refer (if sometimes critically) to the "Pittsburgh Model of Function". So, I think this can be published as is, though I have these residual quibbles.

Thank you very much!

1) They say, of the top level in that model, that.…"Evolutionary Implications can refer to selection upon a trait driven by the object in the past, as in selected effect (Millikan, 1989), but it can equally describe novel adaptive effects, as one may expect in gene birth." Not sure what an "adaptive effect" is. If selection is a force and the effect is fitness-enhancing in environment E, then from the very instant that organism O finds itself in E, this "adaptive effect" is under selection, though possibly only purifying selection if it was previously fixed in another environment, where it had no, or another, effect on fitness. If selection is an outcome, then nothing is a selected effect unless actual mutations have actually been eliminated by purifying selection just yesterday, which seems silly. Any species emerging from a bottleneck has no functions, by that account.

As the reviewer correctly understood, by “adaptive effect” we meant a positive, fitness-enhancing effect. We have clarified the sentence accordingly.

This sentence was meant to clarify that “evolutionary implications” encompasses all the many possible ways an object can influence population dynamics, in contrast with the strict “selected effect” defined in Millikan, 1989:

“Putting things very roughly, for an item A to have a function F as a "proper function", it is necessary (and close to sufficient) that one of these two conditions should hold. (1) A originated as a "reproduction" (to give one example, as a copy, or a copy of a copy) of some prior item or items that, due in part to possession of the properties reproduced, have actually performed F in the past, and A exists because (causally historically because) of this or these performances. (2) A originated as the product of some prior device that, given its circumstances, had performance of F as a proper function and that, under those circumstances, normally causes F to be performed by means of producing an item like A.”

Novel fitness enhancing effects are not included in this strict definition since they did not *cause* the existence of the object. However, our manuscript did not clearly explain that “selected effect” is more restricted that the broad notion of “selection”. This is now clarified also in our re-revised manuscript.

Modifications included in our re-revised manuscript are listed below:

“For most evolutionary biologists, function relates to selection, i.e. the effect for which the gene was selected in the past at the organismal level.”

“In brief, the conversation divides those who argue function should mean why an entity does what it does (the selected effect definition of function) and those who argue it can also mean what an entity does (the causal role definition of function) (Laubichler et al., 2015). Function strictly defined as selected effect is the historical explanation for the existence of an entity (Millikan, 1989). Function as causal role is ahistorical and describes the contribution of an entity to a complex system (Cummins, 1975).”

“The molecular objects of study are thus transitioning between a state without a function and a state with a function. They cannot have a function by the strict selected effect definition (Millikan, 1989), since their existence cannot be explained historically by a past selection.”

“What if the protein happens to confer a fitness benefit to the organism? Still, the locus would not have a function according to the strict selected effect definition (Millikan, 1989), although it might be in the process of acquiring one.”

“For instance, Evolutionary Implications can refer to selection upon a trait driven by the object in the past, as in the strict definition of selected effect (Millikan, 1989), but it can equally describe other ways the object may influence population dynamics such as runaway selection or novel fitness enhancing effects.”

2) The statement "For a genomic sequence to be labelled as a gene, it must by definition have a function." needs some further elaboration or justification (as in the response to reviewers.

We have further elaborated:

“For a genomic sequence to be labelled as a gene, it must by definition have a function; it must express a product that participates in cellular processes and affect phenotypes in a way that is being maintained by selection.”

3) And to say "The molecular objects of study are thus transitioning between a state without a function and a state with a function. They cannot have a function by the selected effect definition, since their existence cannot be explained by a past selection." seems a little problematic. If by a "new gene" one means a gene that was never detected in any other species, but it is fixed in the species in which it is detected, and that species has a reasonable population size, then it likely was selected for, in that species, even if we do not know why. And I suspect that most of the literature on de novo genes describes such situations, not situations in which the "new gene" is a rare polymorphism in the species in which it is described. So it's really only the parent of the new gene that is not a gene: at the very instant that a replicate of this non-gene parent starts to increase in frequency in its population because of a fitness advantage to its bearers, it has acquired 'function' and becomes a bona fide gene, I think. I don't see this as a gradual process or transition, which the authors in their next paragraph, seem to imply. Of course, this could lead further into a discussion of whether selection is a force or an outcome or whatever, and what does fitness mean anyway, but I don't think authors need to go there to be clearer on this point.

As in our response to the first comment, we were referring to the strict selected effect definition of Millikan, 1989 rather than the broader notion of selection. It is also noteworthy that not all loci that are studied in the field of de novo gene birth are fixed within a species. That said, the reviewer’s definition of a de novo gene is absolutely correct under a definition of function as selection, but what is then the “non-gene” parent of the de novo gene? If the new gene starts increasing in frequency, it must be because of something its product does to the organism. What is a locus that is expressed, whose product “does something”, but that is not a gene and whose existence has not been caused by a past selection? Can it be lumped with other “non-gene” sequences that are not expressed at all? We believe these are valid questions but, just like the question of whether selection is a force or an outcome, they fall beyond the scope of our manuscript. We have clarified this sentence by highlighting that the problem lies in the moment of transition from non-gene to gene, when a past selection cannot be invoked:

“The molecular objects of study are thus transitioning between a state without a function and a state with a function. They cannot have upon birth a function by the strict selected effect definition (Millikan, 1989), since their existence cannot have been caused historically by a past selection. The transformative nature of the de novo emergence process thus renders the debates about when and how a locus actually becomes functional highly contentious”

Reviewer #3:The authors have greatly improved the quality of the manuscript and correctly respond to the majority of my comments. There is however 2 points for which the response given by the authors let me unsatisfied.

Thank you very much for your comments which we agree have greatly benefited our manuscript. We have modified it further to include additional clarifications based on your remaining two comments.

1) The first one is about the hierarchical relationships introduced by the authors in their model, and symbolized by ascendant arrows in table 1. Since, as acknowledged by the authors, these relationships are a "surplus" meaning injected in, and not extracted from, the rhetorical analysis (so the model is not purely data-driven), the question remains of where this surplus meaning comes from, and why is it needed. As I have said in my initial comment, this view of hierarchically organized relationships is not new, and corresponds to the classical bottom-up causal relationships from lower (gene) to higher level of organization. This is not a question of logic, but a question of causal relationship between different levels of properties, similar to the classical causal relationship usually expressed by arrows from gene to proteins to pathways to subcellular mechanisms to cells to tissues to organs, etc. (see for example Noble, 2006). Saying so, I do not mean that the model is irrelevant, but that, since the authors privilege a bottom-up view of causal relationships between properties expressed at different level of organization which is not consensual, they should be more explicit about the import of this causal pattern in their model (including bibliographical references) and its justification. Also, some expressions seem misleading or inappropriate. For example, I don't see what is a "logical flow of genetic information" from lower to upper levels of organizational properties, and top-down causal relationships also exist (typically, gene expression is a consequence of interactions). The notion of "spatiotemporal relationship between concepts" does not seem to me relevant. First, the spatiotemporal relationships to which the authors refer are between properties of biological entities, not between concepts, and, in the case of de novo gene, the spatiotemporal relationship between interactions and expression is primarily from interaction to expression, not the opposite, since the expression of a de novo gene cannot occur without prior interaction of the DNA segment with a lot of different proteins involved in a complex interaction network.If we admit the relevance of the directional necessary-but-not sufficient causal relationship between from Expression to Evolutionary implications, it can be logically inferred that each level of meaning of function is inclusive of all the meanings of lower levels. The authors should be more explicit in which way this modifies a model of function, compared to a model without these hierarchical relationships (basically, they should explain why they have chosen to add this hierarchical relationship and how it improves the model). Since previous studies have already identified different notions of functions (see for example Wouters, "Four notions of biological functions", 2003), the authors should refer to these previous studies in their discussion.

We thank the reviewer for these explanations, which clarified for us why the reviewer feels our model is not new in his views. We argue that our model is, in fact, new, because “function” has not been modeled this way before. But it is indeed grounded in a bottom-up understanding of molecular biology that is not itself new. We have added these much-needed clarifications in our re-revised manuscript, along with additional citations and modifications of the expressions that the reviewer had issues with. Furthermore, we remove mentions of the directional necessary-but-not-sufficient relationships between meanings which caused the reviewer’s confusion, and clarify that our model is the way it is because it is field-specific and may not extend to other fields. Changes to the manuscript are listed below:

“We sought to construct an understanding of function specifically tailored to de novo gene birth. We reasoned that this aim would be best achieved by studying how the term is used in the scientific practice of this particular field of research. Indeed, the objects of study and the technical methodologies in this field may lend themselves to different interpretations of function than in other fields such as regulatory genomics, physiology or ecology.”

“Conveniently, these five meanings map to an interpretation of the epistemological flow of genetic information over time and space. Starting from an object’s presence (Expression), we consider its physical properties (Capacities), binding partners within a system (Interactions), phenotypic impact (Physiological Implications) and influence on population dynamics (Evolutionary Implications). Accordingly, we propose to relate these five meanings of function in a hierarchy inspired from molecular, evolutionary and systems biology (Noble, 2006; Medina, 2005; Ernst and Carvunis, 2018). This hierarchy reflects a possible ordering of the series of properties that must be acquired by a locus to undergo de novo gene birth. Altogether, the definitions and hierarchical organization are hereafter referred to as the Pittsburgh model of function (Table 1).”

“Table 1: The Pittsburgh model of function. The hierarchical order of the meanings did not directly derive from our textual analysis, but was inspired from a reductionist interpretation of the flow of genetic information over time and space. It also reflects a possible ordering of the series of properties that must be acquired by a locus to undergo de novo gene birth.”

“Like the molecules they describe, the five meanings of function are interrelated in complex bottom-up and top-down ways that complicate causal inferences (Noble, 2006). For instance, as has been discussed in the context of the ENCODE controversy, Expression is not sufficient to cause Evolutionary Implications (Doolittle, 2018). Inversely, Evolutionary Implications do not necessarily imply Expression since a locus can influence population dynamics through a DNA regulatory activity. The methodological details of the study determine whether the burden of proof has been met to assign one or several of the proposed five meanings of function to a molecular object. Such rigor in functional inference is especially critical for the field of de novo gene birth, where the objects of interest often display some but not all of the properties of established genes (McLysaght and Hurst, 2016; Carvunis et al., 2012; Ruiz-Orera et al., 2018). Our model acknowledges epistemological relationships between different meanings of function while enabling researchers to describe them independently of each other.”

“Other theories of biological function have been proposed (Wouters, 2003; Mossio et al., 2009; Roux, 2014)”

“This model presents practical advantages relative to pre-existing ones because it is tailored to one field of research. In particular, it differentiates between different types of biochemical activities for de novo emerging sequences; this enables scientists to articulate more specific functional inferences than with broad terms that generalize across fields such as causal role or mere activities (Cummins,, 1975; Wouters, 2003; Doolitle, 2013). Rather than theorizing upon the legitimacy of what function should mean, our model decomposes this complex concept into a hierarchical organization of measurable properties of the object in time and space according to meanings that are circulating in the field currently. It can be seen as a conceptual tool well adapted to describe molecular objects (DNA loci, RNA transcripts and proteins) in a manner that roughly maps to subfields of training and associated measurement techniques currently used in de novo gene birth research.”

“There may be additional meanings available beyond the five we included, and different ways to characterize the relationships between these meanings. Function can for instance refer to an object’s influence on ecological behavior, but this was not the case in the small sample of abstracts we analyzed. Interactions may be better placed below Expression in a hierarchical model that would be tailored to regulatory genomics, where the focus is on how physical interaction of DNA elements with diverse proteins determine regulatory outputs, rather than for de novo gene birth, where the focus is on the loci being expressed themselves.”

2) The second point is about the claim, which is also a section of the article, that "The multiple meanings of function hinder scientific communication in the field of de novo gene birth". As I have previously explained, I don't think that, from a biologist viewpoint, the notion of function is confusing, and that is impairs the scientific communication. However, the point is not that I disagree with the authors, but is that the study of the authors provides no evidence for such a claim. The only factual evidence is that, in a set of four encoders, among whom one is an expert is the field, one non-consensual case remains on the 20 abstracts analyzed. Due to the numerous limitations of the study, both in its conception and methodology, these results are clearly unsubstantial; also, non-consensus and discrepancies between members can be interpreted as the inability of the so-called Pittsburg model to account for the actual use of the concept by the researchers themselves. Due to all these limitations, I find such a claim, and the final recommendations, an overinterpretation of the results and, in my opinion, should be removed, or at least reformulated in a less affirmative way.

We agree that our small sample size prevents us from making strong claims, and we have reformulated our manuscript accordingly. We also clarify on what bases we conclude that function is difficult to interpret in the field of de novo gene birth, and that the Pittsburgh model is adequate in this context.

We clarify at the end of the Introduction that our work is grounded on an analysis of how scientists use the word function:

” All these challenges make de novogene evolution a well-suited test bed to evaluate what meanings of function are circulating in this field and whether, and to what extent, the meanings of function hinder scientific communication.”

The title of this section is changed to: How multiple meanings of function are used in the field of de novo gene birth instead of our original claim.

We toned down our conclusion:

“This analysis indicates that when the meaning of function is unspecified, the literature in this field of research can become confusing.”

- Yet our rhetorical approach leads us to conclude that the use of function in this field is hard to interpret, and that further nuance in writing would assist the reader in understanding how the authors intend their results to be interpreted.

“Our rhetorical analysis shows that function becomes an ambiguous concept when applied to the edge case of de novo gene birth, which can lead to confusion in interpretation of the literature. Whether multiple meanings are intended by the authors, or meanings unintended by the authors are interpreted by the readers, or a combination of both, the fact is that scientific communication is hindered by the use of this word within the de novo gene birth field. There may be excellent theoretical arguments to be made about why function *should* mean one thing or another, but the cultural diversity of readers in this emerging field effectively prevents a unique definition to be imposed in practice. We believe that it will be productive for scientists to acknowledge the diversity of onto-epistemological perspectives in this field and adapt their writing style accordingly. Rather than privileging one meaning of function over another, we endorse qualifying the use of function or avoiding the word altogether (Doolittle, 2018). We hope our model will provide a useful tool for scientists to contextualize their writing so the relationship between the observations reported and the functional inferences made can be clarified and the risk of misunderstanding can be reduced.”

We explicitly state why our model appears to be adequate by rephrasing two portions of the second paragraph:

“The abstracts generally provided enough context for each independent reader to confidently assign meanings to most instances, with only rare assignments of the label “vague” (9 instances with one or more vague assignment, none unanimous). Hence, the Pittsburgh model gave readers a key to decipher more specifically what authors meant by the word function.”

“Consensus was successfully reached for all but one instance, suggesting that the Pittsburgh model enables adequate descriptions of most of the instances of function in our abstract database.”

The reasoning is also encapsulated in a newly written narrative legend for Figure 1 in response to an additional comment from the editor:

“Figure 1: Interpreting the word function in scientific abstracts related to de novo gene birth. We analyzed a sample of 20 abstracts containing 42 instances where the word function or one of its derivatives was used to describe DNA, RNA or protein objects. First, each of us read the abstracts independently and assigned one or several of the meanings of function defined in the Pittsburgh model to each instance of function. The distribution of the number of distinct meanings that we assigned to the 42 instances is shown in panel A. For only 5 instances did all of us independently assign the same unique meaning, suggesting that function is most often interpreted in multiple ways by independent readers. Next, we discussed each instance to see if we could reach consensus assignments based on the textual evidence. Consensus was built through conversations and agreement between the readers, rather than majority opinion. The distribution of the number of unique meanings assigned after consensus agreement to each of the 42 instances is shown in panel B. Most (26/42) instances are now assigned to a single meaning. When more than one meaning remains, the readers agreed that the textual evidence supported multiple meanings except for one instance where consensus could not be reached and three meanings were assigned to reflect all the differing interpretations of our team members. In panel C, we show the number of times each of the five meanings of function defined in the Pittsburgh model is assigned to an instance of function.”